# On the Private Estimation of Smooth Transport Maps

**Clément Lalanne** [1]  **Franck Iutzeler** [1]  **Jean-Michel Loubes** [2] [1]  **Julien Chhor** [3]

## Abstract

Estimating optimal transport maps between two distributions from respective samples is an important element for many machine learning methods. To do so, rather than extending discrete transport maps, it has been shown that estimating the Brenier potential of the transport problem and obtaining a transport map through its gradient is near minimax optimal for smooth problems. In this paper, we investigate the private estimation of such potentials and transport maps with respect to the distribution samples. We propose a differentially private transport map estimator achieving an $L^2$ error of at most $n^{-1} \vee n^{-\frac{2\alpha}{2\alpha-2+d}} \vee (n\epsilon)^{-\frac{2\alpha}{2\alpha+d}}$ up to poly-logarithmic terms where $n$ is the sample size, $\epsilon$ is the desired level of privacy, $\alpha$ is the smoothness of the true transport map, and $d$ is the dimension of the feature space. We also provide a lower bound for the problem.

## 1. Introduction

Given two probability measures $P, Q$ on $\mathbb{R}^d$, the question of "how to optimally move the mass" from $P$ to $Q$, i.e. to find a map $T_0 : \mathbb{R}^d \to \mathbb{R}^d$ that solves the Monge (Monge, 1781) problem

$$T_0 \in \operatorname*{argmin} \int_{\mathbb{R}^d} \|T(x) - x\|^2 dP(x) \tag{1}$$
$$\text{s.t.} \quad T_\# P = Q$$

where $T_\# P$ denotes the *push-forward* of $P$ by $T$ is of high interest in both theoretical and more numerical branches of mathematics. The problem is referred to as an *Optimal Transport Problem*, and the optimal mapping $T_0$ is referred to as the *Optimal Transport Map*; we refer the reader to the

textbooks (Villani, 2003; Villani et al., 2009; Santambrogio, 2016; Peyré & Cuturi, 2019) for a general introduction to optimal transport and its computational aspects.

Because of its utility in measuring geometrical discrepancies between measures, and because of recent algorithmic developments (Cuturi, 2013; Altschuler et al., 2017; Dvurechensky et al., 2018), it has become a standard tool in Computer Science (Feydy et al., 2017; Lavenant et al., 2018; Solomon et al., 2015; 2016), Machine Learning (Alaux et al., 2018; Alvarez-Melis et al., 2018; Arjovsky et al., 2017; Cañas & Rosasco, 2012; Gordaliza et al., 2019; Flamary et al., 2018; Genevay et al., 2018; Grave et al., 2019; Janati et al., 2019; Montavon et al., 2016; Schmitz et al., 2018; Staib et al., 2017; Le et al., 2024) and Statistics (del Barrio et al., 2024; Del Barrio et al., 2024; Cazelles et al., 2018; del Barrio et al., 2019; Klatt et al., 2020; Kroshnin et al., 2019; Panaretos & Zemel, 2019; Ramdas et al., 2017; Rigollet & Weed, 2018; Seguy & Cuturi, 2015; Tameling & Munk, 2018; Weed & Berthet, 2019; Zemel & Panaretos, 2019).

When dealing with real-world data, the true distributions $P$ and $Q$ are often accessible only through samples (Courty et al., 2017a; 2014; 2017b; Damodaran et al., 2018; Forrow et al., 2019; Perrot et al., 2016; Seguy et al., 2018; Hütter & Rigollet, 2021). In this article, we suppose that we have access to

$$X_1, \ldots, X_n \overset{\text{i.i.d.}}{\sim} P \quad \text{and} \quad Y_1, \ldots, Y_n \overset{\text{i.i.d.}}{\sim} Q$$

such that the $X_i$s and the $Y_i$s are mutually independent.[1] In this scenario, the problem (1) cannot be solved directly to obtain an optimal transport map; instead, it must be estimated using the available samples. To do so, a crude approach would be to replace $P$ and $Q$ with their empirical counterparts. This approach has two main drawbacks: first, it does not specify how to move points across the entire support of $P$; second, it is affected by the curse of dimensionality (Niles-Weed & Rigollet, 2022). To resolve these issues, (Hütter & Rigollet, 2021) proposed incorporating smoothness and regularity assumptions into the optimal transport map $T_0$ and leveraging functional estimators.

---

[1]Institut de Mathématiques de Toulouse, UMR5219, Université de Toulouse, CNRS, UPS, F-31062 Toulouse Cedex 9, France [2]INRIA, France [3]Toulouse School of Economics, Université Toulouse Capitole, France. Correspondence to: Clément Lalanne <clement.lalanne@math.univ-toulouse.fr>.

*Proceedings of the 42$^{nd}$ International Conference on Machine Learning*, Vancouver, Canada. PMLR 267, 2025. Copyright 2025 by the author(s).

---

[1]For simplicity, we take the same number of samples for both distributions. Our results naturally extend to the case where the two sample sizes are different, at the cost of using more involved notation.

At the same time, deriving estimators from real user data raises new challenges, especially regarding privacy. It is well-documented that sharing statistics based on such data without adequate protections can lead to serious privacy leakages (Narayanan & Shmatikov, 2006; Backstrom et al., 2007; Fredrikson et al., 2015; Dinur & Nissim, 2003; Homer et al., 2008; Loukides et al., 2010; Narayanan & Shmatikov, 2008; Sweeney, 2000; Wagner & Eckhoff, 2018; Sweeney, 2002).

In the context of transport maps estimation, ensuring privacy is highly beneficial for applications involving infringement of fundamental rights as in bias analysis of machine learning algorithms. For instance, in (De Lara et al., 2024) or (Black et al., 2020), optimal transport maps enable to build counterfactual individuals to assess whether an algorithm is prone to discrimination and to mitigate its effect.

To mitigate these concerns, differential privacy (DP) (Dwork et al., 2006) has become the benchmark to ensure privacy protection. DP introduces randomness into computations, ensuring that the released statistics are determined not solely by the dataset but also by the added randomness. This limits the influence of any single data point on the outcome, thereby preserving privacy. Major organizations, including the US Census Bureau (Abowd, 2018), Google (Erlingsson et al., 2014), Apple (Thakurta et al., 2017), and Microsoft (Ding et al., 2017), have embraced this methodology.

This work investigates the problem of estimating smooth optimal transport maps from samples under differentially privacy. In particular, we prove that the optimal estimation rate is expected to degrade under specific regimes.

## 1.1. Contributions

The main contributions of this article can be summarized as follows:

- **A Differentially Private Estimator for Transport Maps.** We introduce a private mechanism (under so-called pure differential privacy) which is aimed at solving the problem of smooth optimal transport maps estimation (see Section 3). This estimator is then adapted to be implementable in practice in Section 6.

- **Statistical Upper Bound.** We analyze this estimator in Section 4 and provide an upper bound on its convergence rate.

- **Minimax Lower Bound.** We provide in Section 5 a minimax lower bound on the private optimal transport map problem in order to characterize the difficulty of the problem.

## 1.2. Related Work

**Differential Privacy and Statistics/Learning.** Over the past decade, there has been growing interest in estimating various quantities while ensuring differential privacy. Examples of relevant works include, but are not limited to, the following references (Wasserman & Zhou, 2010; Barber & Duchi, 2014; Diakonikolas et al., 2015; Karwa & Vadhan, 2018; Bun et al., 2019; 2021; Kamath et al., 2019; Biswas et al., 2020; Kamath et al., 2020; Acharya et al., 2021; Lalanne, 2023; Aden-Ali et al., 2021; Cai et al., 2019; Brown et al., 2021; Cai et al., 2019; Kamath et al., 2022a; Lalanne et al., 2023c;d; Lalanne & Gadat, 2024; Singhal, 2023; Kamath et al., 2023; 2022b).

More specifically, this article falls into the scope of private nonparametric statistics (Tsybakov, 2009), where the quantities of interest live in *infinite* dimensional vector spaces, necessitating the use of approximation techniques alongside estimation. The problem of non-parametric density estimation under differential privacy has been studied in various works (Wasserman & Zhou, 2010; Barber & Duchi, 2014; Lalanne et al., 2023b; Lalanne & Gadat, 2024), both in the setting of this article and in the setup of local privacy (without a trusted server) (Evfimievski et al., 2003; Kasiviswanathan et al., 2008; Duchi et al., 2013; 2016; Butucea et al., 2019; Kroll, 2021; Schluttenhofer & Johannes, 2022; Györfi & Kroll, 2023). Additional works include studies on nonparametric regression (Berrett et al., 2021; Györfi & Kroll, 2022), nonparametric tests (Lam-Weil et al., 2022), interactions between robustness and privacy (Chhor & Sentenac, 2023) and recent advances in locally private Bayesian modeling (Beraha et al., 2023).

To the best of our knowledge, our work is the first to address private estimation of smooth transport maps with statistical guarantees. A noteworthy exception is the article (Xian et al., 2024) which uses elements of optimal transport in a private way in order to obtain fairness with statistical guarantees. Their approach is discussed in the next paragraph.

**Differential Privacy and Optimal Transport** A line of work (Rakotomamonjy & Ralaivola, 2021), building on the ideas from (Harder et al., 2021) provides privacy guarantees for the values of the sliced Wasserstein distance and MMD, respectively.

On another note, other research has explored task-specific private methods utilizing optimal transport. For instance, (Sebag et al., 2023) employs the sliced Wasserstein distance for data generation using an approach based on gradient flows. Additionally, (Tien et al., 2019) addresses differentially private domain adaptation via optimal transport by perturbing the optimal coupling between noisy datasets.

More recently, (Xian et al., 2024) investigated fair and pri-

vate regression, using tools from optimal transport theory. More precisely, they build private histogram estimators of the distributions $P$ and $Q$ and then compute the maps to their Wasserstein barycenter (which has applications in fairness). Their work is not directly related to our study, since the authors focus on fairness guarantees rather than on the problem of learning transport maps.

Beyond these efforts, optimal transport has also been explored within novel privacy paradigms outside the scope of our work (Pierquin et al., 2024; Kawamoto & Murakami, 2019; Yang et al., 2024).

### 1.3. Notation

The symbols $\mathbb{N}$, $\mathbb{Z}$, $\mathbb{R}$ and $\mathbb{C}$ are respectively used to refer to the sets of natural numbers (including 0), relative numbers, real numbers, and complex numbers. For any $k \in \mathbb{N}$, $\mathcal{C}^k(\mathcal{S})$ denotes the set of functions from a space $\mathcal{S}$ to $\mathbb{C}$ that are $k$ times continuously differentiable, and $\mathcal{C}^\infty(\mathcal{S})$ is defined as $\cap_{k \in \mathbb{N}} \mathcal{C}^k(\mathcal{S})$. Whenever applicable, $\nabla$ and $\nabla^2$ are used to refer to gradient and Hessian operators, respectively. For a subset $S$ of a normed vector space, $|S|$ refers to $\sup_{x \in S} \|x\|$ for the inherited norm. For any set $S$, $\#(S)$ denotes its cardinality. For a family $F$ of vectors, $\mathrm{Span}(F)$ refers to the smallest vector space containing $F$. For a closed convex set of an Hilbert space, $\mathrm{Proj}_C$ refers to the convex projection onto $C$. The Hölder norm of order $\alpha$ on $S$ (see Appendix B in (Hütter & Rigollet, 2021)) is denoted by $\|T\|_{C^\alpha(S)}$. For a measure $\mu$, $\| \cdot \|_{L^2(\mu)}$ is the usual $L^2$ norm with reference measure $\mu$. With a small overlap of notation, for a measurable $S \subset \mathbb{R}^d$, $\|\cdot\|_{L^2(S)}$ refers to the $L^2$ norm with reference measure $\mathbb{1}_S \cdot \lambda$ where $\lambda$ is Lebesgue's measure. The asymptotic regimes are considered when $n \to +\infty$ and $n\epsilon \to +\infty$.

## 2. Smooth Transport Maps Estimation

In this section, we cover some basic theory on optimal transport and the minimax estimation of smooth transport maps. Following (Hütter & Rigollet, 2021), we set the feature space[2] $\Omega = [0, 1]^d$ and define $\tilde{\Omega} = [-1, 2]^d$. The notation $M$ (resp. $R$) will refer to a positive constant strictly greater than 2 (resp. $\|$ Identity $\|_{C^\alpha(\tilde{\Omega})}$) throughout the paper. The constants will hide terms that depend on $\Omega$, $\tilde{\Omega}$, $M$, $\alpha$ and $d$, which was also the case in (Hütter & Rigollet, 2021).

### 2.1. Semi-Dual Problem and Smoothness

In the case where $P$ and $Q$ are supported in $\Omega$ and are absolutely continuous with respect to Lebesgue's measure,

Brenier's theorem (Brenier, 1991) ensures that the solution to Problem (1) is a transport map $T_0 = \nabla f_0$ where $f_0$ is a convex function. Furthermore, $f_0$ can be equivalently obtained by solving the semi-dual problem

$$f_0 \in \operatorname*{argmin} \underbrace{\int f(x)dP(x) + \int f^*(y)dQ(y)}_{=:S(f)} \tag{2}$$
$$\text{s.t.} \quad f \in L^1(P)$$

where for all $y$

$$f^*(y) = \sup_{x \in \tilde{\Omega}} \langle x, y \rangle - f(x) \tag{3}$$

is the Fenchel-Legendre transform of $f$;[3] see e.g. (Rüschendorf & Rachev, 1990). In this problem, $f$ is called the *(Brenier) potential*.

In order to estimate transport maps through Brenier potentials and obtain convergence rates, we need to enforce some regularity conditions, which we detail below. First, we require the first probability distribution to be absolutely continuous with lower and upper bounded density.

**Definition 2.1** (Admissible source distributions). We denote by $\mathcal{M}$ the set of probability measures $P$ on $\mathbb{R}^d$ that are supported on $\Omega$, that are absolutely continuous w.r.t. Lebesgue's measure and whose density $\rho_P$ verifies $\frac{1}{M} \leq \rho_P(x) \leq M$ for almost all $x$ in $\Omega$.

Furthermore, we control the bias by imposing smoothness assumptions on the optimal transport map. The following definition introduces the class of admissible smooth transport maps.

**Definition 2.2** (Admissible smooth transport maps). We denote by $\mathcal{T}$ the set of differentiable mappings $T : \tilde{\Omega} \to \mathbb{R}^d$ such that $T = \nabla f$ for some differentiable convex function $f : \tilde{\Omega} \to \mathbb{R}^d$ and

- $\forall x \in \tilde{\Omega}, \quad \|T(x)\| \leq M$,
- $\forall x \in \tilde{\Omega}, \frac{1}{M} \preceq \nabla^2 f(x) \preceq M$,
- $P_{\#T}(\Omega) = 1$.

Furthermore, for $\alpha > 1$ and $R > 1$, we define

$$\mathcal{T}_\alpha(R) = \left\{ T \in \mathcal{T} : \begin{cases} T \text{ is } \lfloor \alpha \rfloor \text{ times differentiable,} \\ \text{and } \|T\|_{C^\alpha(\tilde{\Omega})} \leq R \end{cases} \right\}. \tag{4}$$

In words, admissible transport maps have to be bounded (first item), come from a smooth and strongly convex potential (second item), and have a stable support (third item). In

---

[2]As in (Hütter & Rigollet, 2021), $\Omega$ can be replaced by any bounded and connected Lipschitz domain. In this case, $\tilde{\Omega}$ can be replaced by an hypercube containing $\Omega$ in its interior.

[3]We restrict the sup to $\tilde{\Omega}$ because we are considering extended functions as in Appendix A in (Hütter & Rigollet, 2021).

addition, we will assume that the optimal transport map belongs to the set $\mathcal{T}_\alpha(R)$ of admissible transport maps whose Hölder norm of order $\alpha$ is bounded by $R$.

## 2.2. Empirical Semi-Dual

Hütter & Rigollet (2021) observed that the objective in (2) consists of the sum of an expectation over $P$ and an expectation over $Q$, which can thus be approximated using Monte-Carlo averages. This led them to propose an estimator of the form

$$\hat{T}_0 := \nabla \hat{f}_0 , \qquad (5)$$

where

$$\hat{f}_0 \in \arg\min \quad \underbrace{\frac{1}{n}\sum_{i=1}^n f(X_i) + \frac{1}{n}\sum_{i=1}^n f^*(Y_i)}_{=:\hat{S}(f|X_{1:n},Y_{1:n})} \qquad (6)$$
$$\text{s.t.} \quad f \in \hat{V}_0$$

and where $\hat{V}_0 \subset L^2(\mathbb{R}^d)$ is a functional space to be specified later. Note that $S$ and $\hat{S}$ are linked by the relation

$$\forall f, \quad S(f) = \mathbb{E}_{X_{1:n},Y_{1:n}}\left( \hat{S}(f|X_{1:n},Y_{1:n}) \right) . \quad (7)$$

Controlling the deviations of $\hat{S}$ from its expectation, as well as using $S(f) - S(f_0)$ effectively as a for the transport maps suboptimality, typically requires regularity of the admissible potentials. This is formalized in the following definition mirroring Definition 2.2.

**Definition 2.3** (Admissible potentials). We denote by $\mathcal{X}(M)$ the set of twice continuously differentiable functions $f : \tilde{\Omega} \to \mathbb{R}$ such that $\forall x \in \tilde{\Omega}$

- $|f(x)| \leq 2M^2$,

- $\|\nabla f(x)\| \leq M$,

- $\frac{1}{M} \preceq \nabla^2 f(x) \preceq M$.

Next, we specify how $\hat{V}_0$ can be chosen to obtain an implementable estimator.

## 2.3. Wavelet Decompositions and Subspace Approximations

In line with standard approaches in nonparametric estimation, and in order to exploit the smoothness of the optimal map, Hütter & Rigollet (2021) suggested using successive approximations over nested subspaces $V_1 \subset V_2 \subset \cdots \subset V_J \subset \cdots \subset L^2(\mathbb{R}^d)$ of $L^2(\mathbb{R}^d)$, where $J \geq 1$ denotes an integer controlling the resolution of the approximation space.

Given a pair $\psi_M$ and $\psi_F$ of smooth enough mother and father wavelets with compact supports, one can define a wavelet Hilbert basis of $L^2(\mathbb{R})$ of the form

$$(\Psi_k^{j,g})_{j\in\mathbb{N},k\in\mathbb{Z}^d,g\in G^j} \qquad (8)$$

where $\forall j$, $G^j$ is finite. The exact construction is presented in Appendix B of (Hütter & Rigollet, 2021), with complements given in the present Appendix A. We do not include it in the main body of this article as we believe that it is not essential to understanding the core message.

Importantly, since we are interested in guarantees in $L^2(\tilde{\Omega})$ rather than $L^2(\mathbb{R}^d)$, and because the supports of the father and mother wavelets are compact, the basis can be restricted to the indices such that $k \in K_j$ for some *finite* $K_j \subset \mathbb{Z}^d$.

Thus, defining

$$V_J := \text{Span}\left( (\Psi_k^{j,g})_{j\leq J,k\in K_j,g\in G^j} \right) \qquad (9)$$

yields a sequence of *finite-dimensional* nested approximation spaces.

The estimator proposed by Hütter & Rigollet (2021) is the solution of (6) where $\hat{V}_0$ is replaced with $V_J \cap \mathcal{X}(2M)$. We denote the solution of the resulting problem and its associated transport map by $\hat{f}_J$ and $\hat{T}_J := \nabla \hat{f}_J$, respectively.

Finally, measuring the *utility* (or error) of a candidate transport map $T$ by its squared $L^2(P)$ distance

$$d(T, T_0)^2 := \int \|T - T_0\|^2 dP , \qquad (10)$$

we can guarantee that such a transport map estimator has a (near) minimax optimal rate.

**Theorem 2.4** (Th. 2 of (Hütter & Rigollet, 2021)). *For any $\alpha > 1$, the minimax rate of smooth optimal transport map estimation is*

$$\inf_{\hat{T}} \sup_{P\in\mathcal{M},T_0\in\mathcal{T}_\alpha} \mathbb{E}_{X_{1:n},Y_{1:n}}\left( d(T,\hat{T}_0)^2 \right) \gtrsim \frac{1}{n} \vee n^{-\frac{2\alpha}{2\alpha-2+d}} . \qquad (11)$$

*Moreover, if $P \in \mathcal{M}$ and $T_0 \in \mathcal{T}_\alpha$, the estimator $\hat{T}_J := \nabla \hat{f}_J$ defined above achieves this rate up to polylogarithmic factors. Here, the constants also hide a dependence on $R$.*

## 2.4. Towards the Analysis of a Private Estimator

In this paper, we extend this approach to the context of differential privacy. We derive a private estimator $\hat{f}_{\text{priv}}$ for the optimal Brenier potential and deduce the corresponding private optimal transport map $\hat{T}_{\text{priv}} = \nabla \hat{f}_{\text{priv}}$. To understand how privacy will affect the utility of the estimation, we propose the following risk decomposition.

**Lemma 2.5.** *Suppose that there exists a random variable* Priv *independent of all the other sources of randomness such that* $\hat{f}_{\text{priv}}$ *is* $(X_{1:n}, Y_{1:n}, \text{Priv})$*-measurable and there exists* $U \geq 0$ *that is* Priv*-measurable such that*

$$\underbrace{\hat{S}(\hat{f}_{\text{priv}}|X_{1:n}, Y_{1:n}) - \hat{S}(\hat{f}_J|X_{1:n}, Y_{1:n})}_{\text{Suboptimality on the semi-dual}} \leq U \qquad (12)$$

*almost surely. If* $\hat{f}_{\text{priv}} \in V_J \cap \mathcal{X}(2M)$ *almost surely, then*

$$\mathbb{E}_{X_{1:n}, Y_{1:n}} \left( d(\hat{T}_{\text{priv}}, T_0)^2 \right)$$
$$\lesssim U + \frac{J 2^{J(d-2)} \ln(1 + Cn)}{n} + \frac{1}{n} \qquad (13)$$
$$+ \inf_{T \in V_J \cap \mathcal{X}(2M)} d(T, T_0)^2$$

Priv*-almost surely for some positive constant* $C$. *Here,* $\mathbb{E}_{X_{1:n}, Y_{1:n}}$ *denotes the expectation w.r.t. the data, i.e. conditional on* Priv.

*Proof.* See Appendix C.1. □

In the next section, we propose a private estimator $\hat{f}_{\text{priv}}$ and explain how to control $\hat{S}(\hat{f}_{\text{priv}}|X_{1:n}, Y_{1:n}) - \hat{S}(\hat{f}_J|X_{1:n}, Y_{1:n})$. In particular, this estimator will satisfy the hypotheses of Lemma 2.5.

# 3. Private Estimator

This section presents some background on differential privacy and introduces our private estimator.

## 3.1. Differential Privacy

Given a (randomized) mechanism $M$ (i.e. a conditional kernel of probabilities), $\text{dom}(M)$ denotes its domain (i.e. the set of admissible inputs) and $\text{codom}(M)$ refers to its codomain (i.e. the set of admissible outputs). The set $\text{dom}(M)$ represents the space of all possible data sets and is equipped with a binary symmetric relation $\cdot \sim \cdot$ called *neighboring* relations. Informally, two datasets are neighboring if they contain the same data except for at most one individual. Formally, we say that $(X_{1:n}, Y_{1:n}) \sim (X'_{1:n}, Y'_{1:n})$ if $d_{\text{ham}}((X_{1:n}, Y_{1:n}), (X'_{1:n}, Y'_{1:n})) \leq 1$, where $d_{\text{ham}}(\cdot, \cdot)$ to denote the Hamming distance,

**Definition 3.1** (Differential Privacy (Dwork et al., 2006)). Given $\epsilon \geq 0$, a randomized mechanism $M : \text{dom}(M) \to \text{codom}(M)$ is $\epsilon$-differentially private (or $\epsilon$-DP) if for all $D \sim D' \in \text{dom}(M)$ and all measurable $S \subset \text{codom}(M)$ we have

$$\mathbb{P}_M(M(D) \in S) \leq e^\epsilon \mathbb{P}_M(M(D') \in S) . \qquad (14)$$

Here, $\epsilon \geq 0$ is a parameter that controls the privacy level: Lower values of $\epsilon$ correspond to more private mechanisms $M$.

We will rely on the standard *report noisy max* mechanism (Dwork & Roth, 2014; McSherry & Talwar, 2007; McKenna & Sheldon, 2020; Ding et al., 2021). We give its version instantiated with Laplace noise below.

**Lemma 3.2** (Report Noisy Argmin with Laplace Noise). *Let* $f_1, \ldots, f_N$ *be queries with sensitivities that are uniformly bounded by* $\Delta$, *that is, for any of neighboring datasets* $D \sim D'$ *and for any* $i \in \{1, \ldots, n\}$,

$$|f_i(D) - f_i(D')| \leq \Delta . \qquad (15)$$

*Then, if* $L_1, \ldots, L_N$ *are independent and identically distributed random variables with standard Laplace distribution, the mechanism* $\hat{i}$ *defined as*

$$\hat{i}(D) \in \text{argmin} \left\{ f_i(D) + \frac{2\Delta}{\epsilon} L_i \right\} \qquad (16)$$

*is* $\epsilon$-DP, *with the convention that we return a random index in the* argmin *when it is not unique.*

In particular, we will use this mechanism as a building block to construct a private estimator of $\hat{f}_J$.

## 3.2. Noisy semi-dual estimator

In this section, we construct our private estimator, building on the empirical semi-dual one presented above.

The first step is to provide an upper bound on the sensitivity of $\hat{S}(f|X_{1:n}, Y_{1:n})$, which is established in the lemma below.

**Lemma 3.3** (Sensitivity of the semi-dual objective). *For any* $f \in \mathcal{X}(2M)$, *it holds that*

$$\sup_{(X_{1:n}, Y_{1:n}) \sim (X'_{1:n}, Y'_{1:n})} \left| \hat{S}(f|X_{1:n}, Y_{1:n}) - \hat{S}(f|X'_{1:n}, Y'_{1:n}) \right|$$
$$\leq \frac{2\|f\|_\infty \vee 2|\tilde{\Omega}|^2}{n} . \qquad (17)$$

*Proof.* See Appendix D.2. □

Now, recalling Section 2.3, let $C_{J,M}$ be a *finite* subset of $V_J \cap \mathcal{X}(2M)$. We can enumerate this set as

$$C_{J,M} = \{f_1, \ldots, f_{\#(C_{J,M})}\} . \qquad (18)$$

Considering independent random variables $L_1, \ldots, L_{\#(C_{J,M})}$ with Laplace distribution that are also independent of all the other sources of randomness, we define

$$\hat{i}_{\text{priv}}(D) \in \text{argmin} \left\{ \hat{S}(f_i|D) + \frac{32M^2 \vee 36d}{n\epsilon} L_i \right\} \qquad (19)$$
$$\text{for} \quad i \in \{1, \ldots, \#(C_{J,M})\}$$

**Theorem 3.4** (Privacy of the noisy semi-dual estimator).
*The mechanism returning the pair*

$$(\hat{f}_{\text{priv}} := f_{\hat{i}_{\text{priv}}}, \hat{T}_{\text{priv}} := \nabla \hat{f}_{\text{priv}}) \tag{20}$$

*is $\epsilon$-DP.*

*Proof.* It follows as a direct consequence of Lemma 3.2 and of Lemma 3.3 that $\hat{i}_{\text{priv}}$ is $\epsilon$-DP. Hence, $(\hat{f}_{\text{priv}}, \hat{T}_{\text{priv}})$ is $\epsilon$-DP by the post-processing property of differential privacy (Dwork & Roth, 2014). ☐

We will use the estimator defined in Equation (20) as our estimator for both the optimal Brenier potential and the optimal transport map.

A question that remains unanswered in this section is how to choose $J$ and $C_{J,M}$. So far, we have treated these quantities as hyperparameters. The next section explains how to choose them in order to optimize the bias-variance trade-off.

# 4. Statistical Upper Bound

This section details the construction of $C_{J,M}$ and gives an upper bound on the utility of the proposed private estimator.

## 4.1. Covering construction

We now construct the set $C_{J,M}$ such that it forms a $\delta$-*covering* of the space $V_J \cap \mathcal{X}(2M)$ with respect to the empirical criterion $\hat{S}(\cdot|X_{1:n}, Y_{1:n})$. Specifically, we aim to ensure that $C_{J,M} \subset V_J \cap \mathcal{X}(2M)$ and that

$$\forall f \in V_J \cap \mathcal{X}(2M), \exists f_c \in C_{J,M} \text{ s.t.} \\ |\hat{S}(f|X_{1:n}, Y_{1:n}) - \hat{S}(f_c|X_{1:n}, Y_{1:n})| \leq \delta . \tag{21}$$

A first step towards this goal is to observe that $\hat{S}(\cdot|X_{1:n}, Y_{1:n})$ is Lipschitz-continuous with respect to the $L^\infty$ norm, as shown in the following lemma.

**Lemma 4.1.** *For any $f_1, f_2 \in \mathcal{X}(2M)$,*

$$\left| \hat{S}(f_1|X_{1:n}, Y_{1:n}) - \hat{S}(f_2|X_{1:n}, Y_{1:n}) \right| \leq 2\|f_1 - f_2\|_\infty \tag{22}$$

*for any dataset $(X_{1:n}, Y_{1:n})$.*

*Proof.* See Appendix E.1 ☐

Thus, a $\delta > 0$-covering of $V_J \cap \mathcal{X}(2M)$ for $\hat{S}(\cdot|X_{1:n}, Y_{1:n})$ can be built from a $\frac{\delta}{2}$-covering of $V_J \cap \mathcal{X}(2M)$ for $\|\cdot\|_\infty$. The following lemma bounds the cardinality of a specific covering of this form.

**Lemma 4.2.** *There exists a $\delta$-covering of $V_J \cap \mathcal{X}(2M)$ for $\|\cdot\|_\infty$ of cardinality at most $N$ where*

$$\ln(N) \lesssim 2^{Jd} \ln\left(\frac{C2^{Jd/2}}{\delta} + 1\right) \tag{23}$$

*for some constant $C$ that does not depend on $J$ or $\delta$*

*Proof.* See Appendix E.2 ☐

## 4.2. Main result

We now have all the pieces to bound the error of $\hat{T}_{\text{priv}}$. Indeed, we can apply Lemma 2.5 with an lower bound on the empirical semi-dual error ($U$) that depends on (i) the resolution $\delta$ of the covering, and (ii) the uncertainty created by privacy $\max\left\{|L_1|, \dots, |L_{\#(C_{J,M})}|\right\}$. The remaining approximation bias $\inf_{T \in V_J \cap \mathcal{X}(2M)} d(T, T_0)^2$ is controlled as in (Hütter & Rigollet, 2021).

**Theorem 4.3.** *Let $P \in \mathcal{M}, T_0 \in \mathcal{T}_\alpha$. By choosing $\delta$ appropriately, the estimator $\hat{T}_{\text{priv}}$ has a utility satisfying*

$$\mathbb{E}\left(\int \|T_0 - \hat{T}_{\text{priv}}\|^2 dP\right)$$
$$\lesssim R^2 2^{-2J\alpha} + \frac{J2^{J(d-2)}\ln(n)}{n} + \frac{1}{n} + \frac{2^{Jd}\ln(n\epsilon)}{n\epsilon} \tag{24}$$

*for $J$ larger than a sufficiently large constant that also depend on $R$.*

*Proof.* See Appendix E.3 ☐

Ignoring poly-logarithmic terms and treating $R$ as a constant for conciseness, we can optimize over $J$ to obtain an asymptotic upper bound on the error as follows

$$\mathbb{E}\left(\int \|T_0 - \hat{T}_{\text{priv}}\|^2 dP\right) \\ \lesssim n^{-1} \vee n^{-\frac{2\alpha}{2\alpha-2+d}} \vee (n\epsilon)^{-\frac{2\alpha}{2\alpha+d}} . \tag{25}$$

*Remark* 4.4. It is possible to obtain similar bounds that hold with high probability with minimal adaptations to the proofs.

# 5. Lower Bound

Packing arguments from (Hütter & Rigollet, 2021) as well as coupling arguments (Acharya et al., 2018; 2021; Lalanne et al., 2023a) can be applied to handle the DP nature of the constraint, and yield the following lower bound.

**Theorem 5.1** (Lower Bound). *Asymptotically,*

$$\inf_{\hat{T}_{\text{priv}}} \sup_{P \in \mathcal{M}, T_0 \in \mathcal{T}_\alpha} \mathbb{E}\left(\int \|T_0 - \hat{T}_{\text{priv}}\|^2 dP\right) \\ \gtrsim \frac{1}{n} \vee n^{-\frac{2\alpha}{2\alpha-2+d}} \vee (n\epsilon)^{-\frac{2\alpha}{\alpha-1+d}} \tag{26}$$

*where the infimum is taken over all estimators $\hat{T}_{\text{priv}}$'s that are $\epsilon$-DP.*

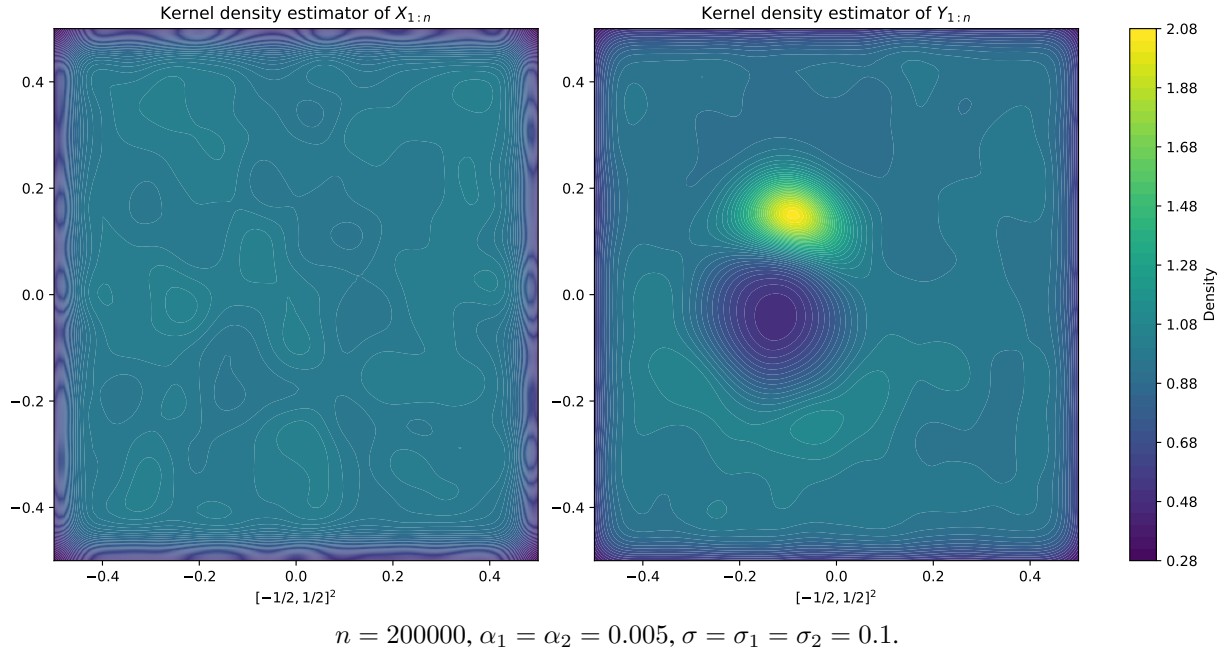

$$n = 200000, \alpha_1 = \alpha_2 = 0.005, \sigma = \sigma_1 = \sigma_2 = 0.1.$$

*Figure 1.* Kernel density estimators of the samples used in the experiments

*Proof.* See Appendix F.1 □

We observe that the term in $n\epsilon$ in the lower bound above does not match the corresponding term in the upper bound (25). It remains unclear which, if any, of these bounds is tight. However, it is worth noting that similar coupling arguments have given optimal lower bounds in other nonparametric problems (Lalanne et al., 2023b; Lalanne & Gadat, 2024). We thus believe that the lower bound is more likely to be optimal than the upper bound.

Following this conjecture, the suboptimality may arise from a suboptimal analysis of the estimator, a suboptimal choice of a covering, or limitations of the proposed estimator itself. Furthermore, our estimator is, to the best of our knowledge, the only one that has been proposed to solve the problem.

Another consequence of this lower bound is that the estimation is provably degraded by privacy (even with the best estimator) when $\epsilon \lesssim n^{-\frac{\alpha-1}{2\alpha-2+d}} \wedge n^{-\frac{\alpha+1-d}{2\alpha}}$.

## 6. Experiments

This section focuses on how to numerically approximate $\hat{f}_{\text{priv}}$ and $\hat{T}_{\text{priv}}$ defined in (20) and presents numerical results to illustrate the proposed method.

### 6.1. Discretization

The main difficulty in computing $\hat{f}_{\text{priv}}$ and $\hat{T}_{\text{priv}}$ is that the Fenchel-Legendre transform $f^*$ of a candidate Brenier po-

tential $f$ may not always have a closed-form expression. To address this issue, (Hütter & Rigollet, 2021) approximate $f^*$ by restricting the supremum in its definition to a grid. This subsection develops similar ideas to adapt this method to the context of differential privacy.

Let $\tilde{\Omega}^{(\text{grid})} \subset \tilde{\Omega}$ be a *finite* approximation set. In practice it is taken to be a uniform grid. We then define

$$f^*_{(\text{grid})}(y) = \sup_{x \in \tilde{\Omega}^{(\text{grid})}} \langle x, y \rangle - f(x), \quad \forall y \in \tilde{\Omega}. \quad (27)$$

Note that the sensitivity analysis in Lemma 3.3 still holds under this approximation of the Fenchel dual.

We represent potentials $f$ by storing their discretization $f_{(\text{grid})}$ on the grid $\tilde{\Omega}^{(\text{grid})}$. Their gradients are approximated using finite differences, denoted by $\nabla_{(\text{grid})} f_{(\text{grid})}$.

Since we now only store the value of the potential and its (approximate) dual on a grid, we need to clip the points to this grid. We thus denote by $X^{(\text{grid})}_{1:n}$ and $Y^{(\text{grid})}_{1:n}$ the modified versions of the datasets $X_{1:n}$ and $Y_{1:n}$ where each point is replaced with its closest neighbor in $\tilde{\Omega}^{(\text{grid})}$.

Finally, if one wishes to apply Lemma 3.3 but cannot guarantee that $f \in \mathcal{X}(2M)$, we propose introducing a clipping constant $C$ in order to safeguard privacy.

This procedure leads to the following practical algorithm: Start with a finite family of candidate discretized potentials

$$F = \left\{ f_{(\text{grid}),1}, \ldots, f_{(\text{grid}),N} \right\}, \quad (28)$$

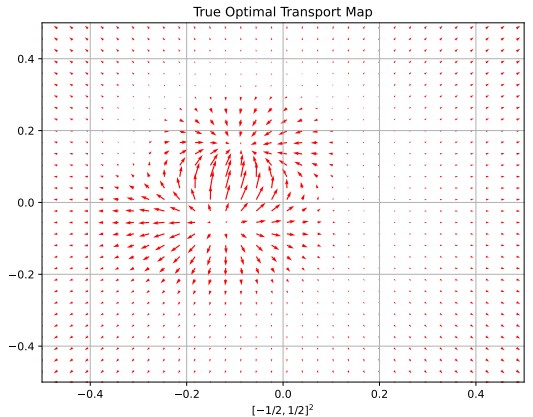
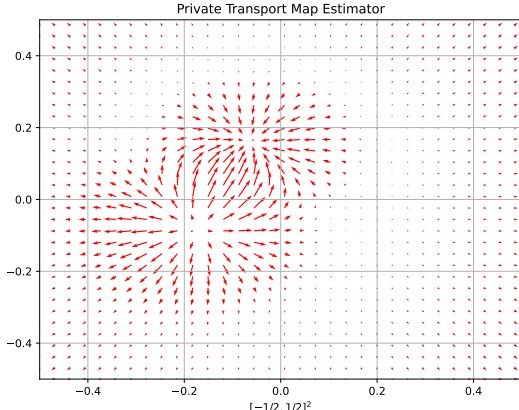

On the left: The true optimal transport map between the underlying distributions in the Gaussian attraction / repulsion model described in Section 6.2. On the right: The transport map that is privately estimated using the approximation algorithm described in Section 6.1. For the numerical values, $\tilde{\Omega}^{(\text{grid})}$ is a uniform grid in $[-1/2, 1/2]^2$ with $64^2$ points, $\epsilon = 1.$, $n = 200000$, $\alpha_1 = \alpha_2 = 0.005$, $\sigma = \sigma_1 = \sigma_2 = 0.1$, $T = 2000$, and $C = 0.25$.

*Figure 2.* Optimal Transport Map VS Estimated Private Map

then compute

$$\hat{i} \in \operatorname{argmin}\left\{\hat{S}^C_{(\text{grid})}(f_{(\text{grid}),i}|X^{(\text{grid})}_{1:n}, Y^{(\text{grid})}_{1:n}) + \frac{4C}{n\epsilon}L_i\right\} \tag{29}$$

where for any $f$, $X_{1:n}$ and $Y_{1:n}$

$$\hat{S}^C_{(\text{grid})}(f|X_{1:n}, Y_{1:n}) := \frac{1}{n}\sum_{i=1}^{n}\operatorname{Proj}_{[-C,C]}f(X_i)$$
$$+ \frac{1}{n}\sum_{i=1}^{n}\operatorname{Proj}_{[-C,C]}f^*_{(\text{grid})}(Y_i) \tag{30}$$

and where $L_1, \dots, L_N$ are independent Laplace random variables. By Lemma 3.2, $\hat{i}$ is $\epsilon$-DP, and by post-processing, so is $\hat{T} := \nabla_{(\text{grid})}f_{(\text{grid}),\hat{i}}$, which we use as an estimator.

## 6.2. Gaussian Attraction / Repulsion Model

We illustrate the behavior of the estimator $\hat{T}$ defined in Section 6.1 on the following model, which we refer to as the Gaussian attraction/repulsion model.

We begin with a simple quadratic potential $f_{\text{quad}}(x) = \frac{1}{2}\|x\|^2$ for which the mapping induced by its gradient is the identity. For any $\mu$ and $\sigma > 0$, denoting the Gaussian potential by $f_{\mathcal{N}}(x|\mu, \sigma) = e^{-\|x-\mu\|^2/(2\sigma^2)}$, we consider potentials of the form

$$f(x|\alpha_1, \alpha_2, \mu_1, \mu_2, \sigma_1, \sigma_2) =$$
$$f_{\text{quad}}(x) + \alpha_1 f_{\mathcal{N}}(x|\mu_1, \sigma_1) - \alpha_2 f_{\mathcal{N}}(x|\mu_2, \sigma_2) \tag{31}$$

where $\alpha_1, \alpha_2, \sigma_1, \sigma_2$ are positive numbers and $\mu_1, \mu_2 \in \mathbb{R}^d$ are location parameters.

Then, a random potential is generated by sampling two independent copies $\mu_1, \mu_2 \sim \mathcal{N}(0, \sigma I_2)$. This defines the "true" potential, which is the target of the estimation.

We then sample $n$ i.i.d. uniform random variables $X_{1:n}$ over $[-1/2, 1/2]^2$, and generate the observations $Y_{1:n}$ by applying the gradient of the previously defined potential to $n$ new i.i.d. uniform random variables over $[-1/2, 1/2]^2$. This is illustrated in Figure 1, where we represent two kernel density estimators constructed on the datasets $X_{1:n}$ and $Y_{1:n}$, respectively.

We then apply the algorithm defined in Section 6.1 where $F$ is a collection of $T$ independent potentials that are generated according to the same distribution as the true potential.

The result of the algorithm and the optimal transport map are represented in Figure 2. This simulation confirms that the estimated transport map is close to the optimal one. The code used to generate those results can be found at /https://github.com/clemlal/PrivateSmoothTransportMapRNArgMin.

## 7. Conclusion

In this paper, we have introduced a private estimator for the problem of privately estimating smooth optimal transport maps. We have analyzed the utility of this estimator and derived a lower bound on the difficulty of the statistical problem. Finally, we proposed an adaptation of this theoretical estimator that is implementable in practice.

This work opens up several research directions. One of them is the quest for an optimal private mechanism, and

another one is the design of numerically efficient algorithms. In both cases, we believe that investigating *regularization* techniques such as *entropic regularization* would be an important step forward as it could help stabilize the problem, reduce the amount of privacy noise, and enable the use of numerically efficient algorithms such as Sinkhorn's algorithm (Cuturi, 2013).

## Acknowledgements

This paper has been partially funded by the Agence Nationale de la Recherche under grants ANR-17-EURE-0010 (Investissements d'Avenir program), ANR-23-CE23-0029 Regul-IA, and ANR-24-CE23-1529 MAD. The authors also acknowledge the support of the AI Cluster ANITI (ANR-19-PI3A-0004). In addition, we thank the anonymous reviewers for the constructive discussions that led, in particular, to the addition of Appendix G and Appendix H.

## Impact Statement

This paper presents work whose goal is to advance the field of Privacy-Preserving Machine Learning. There are many potential societal consequences of our work, none which we feel must be specifically highlighted here.

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

## A. Details on Wavelet Decomposition

In this article, we adopt the same conventions for functional spaces as in (Hütter & Rigollet, 2021). In particular, we encourage the reader to consult Appendix B in (Hütter & Rigollet, 2021) for formal definitions of the functional spaces considered in this paper.

A notable difference from (Hütter & Rigollet, 2021) is that all the functions considered in this work are supported on $\tilde{\Omega}$, which is a hypercube rather than a more general Lipschitz domain. Therefore, we can restrict the wavelet basis to functions whose support is included in $\tilde{\Omega}$ only, thus avoiding overlaps with the complement: for any function in this basis, its support is either included in $\tilde{\Omega}$ or in $\bar{\tilde{\Omega}} \cup \partial\tilde{\Omega}$.

As a consequence, if $f \in V_J$, and if $(\gamma_k^{j,g})_{k,j,g}$ denotes its wavelet coefficients in the corresponding wavelet basis, then

$$\|f\|_{L^2(\tilde{\Omega})}^2 = \|(\gamma_k^{j,g})_{k,j,g}\|_2^2 . \tag{32}$$

## B. Technical Tools

**Lemma B.1.** *There exists an open neighborhood $O$ of $0$ in $\mathbb{R}^{q \times q}$ such that*

$$M \in O \quad \Longrightarrow \quad |\det(I_q + M) - 1 - \operatorname{tr}(M)| \leq \|M\| , \tag{33}$$

*where $O$ is only affected by the choice of $\| \cdot \|$.*

*Proof.* By computing the partial derivatives of $M \mapsto \det(I_q + M)$ in the canonical basis of $\mathbb{R}^{q \times q}$, we immediately see that the gradient of $M \mapsto \det(I_q + M)$ at $0$ for the Euclidean structure induced by the Frobenius inner product is $I_d$. Consequently, since $M \mapsto \det(I_q + M)$ is $\mathcal{C}^1$ (it has a polynomial expression in the coefficients of $M$),

$$\det(I_q + M) = 1 + \operatorname{tr}(M) + o(\|M\|) . \tag{34}$$

The equivalence of norms in finite dimension allows us to conclude. $\square$

**Lemma B.2.** *Let $L_1, \ldots, L_M$ be $M$ independent random variables with Laplace distribution (that is, with pdf $t \mapsto \frac{1}{2}e^{-|t|}$ w.r.t. Lebesgue's measure). Then*

$$\mathbb{E}\left(\max_{i=1,\ldots,N} |L_i|\right) \leq 1 + \ln(N) . \tag{35}$$

*Proof.* Let $\delta \geq 0$, by independence,

$$\begin{aligned}
\mathbb{P}\left(\max_{i=1,\ldots,N} |L_i| \leq \delta\right) &= \mathbb{P}\left(\bigcap_{i=1,\ldots,N} (|L_i| \leq \delta)\right) \\
&= \prod_{i=1}^{N} \mathbb{P}\left((|L_i| \leq \delta)\right) \\
&= \left(1 - e^{-\delta}\right)^N .
\end{aligned} \tag{36}$$

Thus,

$$
\begin{aligned}
\mathbb{E}\left(\max_{i=1,\dots,N}|L_i|\right) &= \int_{[0,+\infty)}\mathbb{P}\left(\max_{i=1,\dots,N}|L_i| > \delta\right)d\delta \\
&= \int_{(0,+\infty)}\left(1 - \left(1 - e^{-\delta}\right)^N\right)d\delta \\
&\stackrel{u=e^{-\delta}}{=} \int_{(0,1)}\left(1 - (1-u)^N\right)\frac{du}{u} \\
&\stackrel{(1-u)^N \geq \max(0,1-Nu)}{=} \int_{(0,1)}\left(1 - \max(0, 1-Nu)\right)\frac{du}{u} \\
&= \int_0^{\frac{1}{N}}\left(1 - \max(0, 1-Nu)\right)\frac{du}{u} + \int_{\frac{1}{N}}^1\left(1 - \max(0, 1-Nu)\right)\frac{du}{u} \\
&\leq 1 + \ln(N) \ .
\end{aligned}
\tag{37}
$$

$\square$

**Lemma B.3.** *Let $f \in V_J$, and denote by $(\gamma_k^{j,g})_{k,j,g}$ its wavelet coefficients for compactly supported mother and father wavelets, and with support adequation to $\tilde{\Omega}$ as described in Appendix A. Then*

$$
\|(\gamma_k^{j,g})_{k,j,g}\|_\infty \leq \|(\gamma_k^{j,g})_{k,j,g}\|_2 \lesssim \|f\|_\infty \lesssim 2^{\frac{Jd}{2}}\|(\gamma_k^{j,g})_{k,j,g}\|_\infty \ .
\tag{38}
$$

*Proof.*

$$
\begin{aligned}
\|f\|_\infty &= \sqrt{\|f^2\|_\infty} \\
&\geq \sqrt{\frac{1}{\operatorname{Vol}\left(\tilde{\Omega}\right)}\int_{\tilde{\Omega}} f^2} \\
&\stackrel{\text{Parseval}}{=} \sqrt{\frac{1}{\operatorname{Vol}\left(\tilde{\Omega}\right)}\|(\gamma_k^{j,g})_{k,j,g}\|_2^2} \\
&\gtrsim \|(\gamma_k^{j,g})_{k,j,g}\|_2 \\
&\geq \|(\gamma_k^{j,g})_{k,j,g}\|_\infty \ .
\end{aligned}
\tag{39}
$$

The inequality

$$
\|f\|_\infty \lesssim 2^{\frac{Jd}{2}}\|(\gamma_k^{j,g})_{k,j,g}\|_\infty
\tag{40}
$$

comes from Lemma 24 in (Hütter & Rigollet, 2021).

$\square$

# C. Proofs of Section 2

## C.1. Proof of Lemma 2.5

Following subsection 5.4 in (Hütter & Rigollet, 2021), we use $\|\cdot\| := \|\cdot\|_{L^2(P)}$ and we define

$$
\forall f, \quad S_0(f) := S(f) - S(f_0), \quad \hat{S}_0(f) := \hat{S}(f|X_{1:n}, Y_{1:n}) - \hat{S}(f_0|X_{1:n}, Y_{1:n}) \ ,
\tag{41}
$$

$$
\forall \tau \geq 0, \quad \mathcal{F}_J(\tau^2) := \{f \in V_J \cap \mathcal{X}(2M) : S_0(f) \leq \tau^2\} \ ,
\tag{42}
$$

and

$$
\bar{f} \in \operatorname{argmin}_{f \in V_J \cap \mathcal{X}(2M)} S_0(f) \ ,
\tag{43}
$$

Furthermore, we define $\hat{f} = \hat{f}_J \in V_J \cap \mathcal{X}(2M)$, and, for $\sigma > 0$, we also let

$$
\hat{f}_s = s\hat{f}_{\text{priv}} + (1-s)\bar{f}, \quad \text{where} \quad s = \frac{\sigma}{\sigma + \|\nabla\hat{f}_{\text{priv}} - \nabla\bar{f}\|} \ .
\tag{44}
$$

Following the same steps as in Proposition 11 from (Hütter & Rigollet, 2021), we obtain

$$\|\nabla \hat{f}_s - \nabla \bar{f}\| = s\|\nabla \hat{f}_{\text{priv}} - \nabla \bar{f}\| = \frac{\sigma \|\nabla \hat{f}_{\text{priv}} - \nabla \bar{f}\|}{\sigma + \|\nabla \hat{f}_{\text{priv}} - \nabla \bar{f}\|} \leq \sigma \tag{45}$$

and

$$S_0(\hat{f}_s) \leq 4M(\sigma^2 + \|\nabla \bar{f} - \nabla f_0\|^2) =: \tau^2 . \tag{46}$$

It follows that, $\hat{f}_s \in \mathcal{F}_J(\tau^2)$ and, therefore, $\bar{f} \in \mathcal{F}_J(\tau^2)$ since we have $S_0(\bar{f}) \leq S_0(\hat{f}_s) \leq \tau^2$ by the definition of $\bar{f}$. We now turn to controlling the error of the semi-dual. We have

$$\begin{aligned}
\hat{S}(\hat{f}_s) &= s\hat{S}(\hat{f}_{\text{priv}}) + (1-s)\hat{S}(\bar{f}) \\
&\leq s\left(\hat{S}(\hat{f}_{\text{priv}}|X_{1:n}, Y_{1:n}) - \hat{S}(\hat{f}|X_{1:n}, Y_{1:n})\right) + s\hat{S}(\hat{f}) + (1-s)\hat{S}(\bar{f}) \\
&\leq U + \hat{S}(\bar{f})
\end{aligned} \tag{47}$$

where the last inequality comes from the optimality of $\hat{f}$ for the empirical semi-dual problem.

Hence,

$$S_0(\hat{f}_s) \leq S_0(\bar{f}) + U + 2 \sup_{f \in \mathcal{F}_J(\tau^2)} |S_0(f) - \hat{S}_0(f)| . \tag{48}$$

The rest of the proof follows the same lines as the end of Section 5.4 in (Hütter & Rigollet, 2021), replacing $S_0(\bar{f})$ with $S_0(\bar{f}) + U$ and $\hat{f}$ with $\hat{f}_{\text{priv}}$.

## D. Proofs of Section 3

### D.1. Proof of Lemma 3.2

This result is folklore in the literature of differential privacy, we include its proof for the sake of completeness. To simplify the notation, let us look at the case where the scaling factor in front of the Laplace random variables in (15) is one. The general case will be obtained by scaling. Let $D \sim D'$ be a pair of neighboring datasets, let $k \in \{1, \ldots, N\}$. We will condition on $(L_{k'})_{k' \neq k}$. All that follow holds almost surely.

$$\mathbb{P}\left(\hat{i}(D) = k | (L_{k'})_{k' \neq k}\right) = \mathbb{P}\left(\cap_{k' \neq k} \left(f_k(D) + L_k < f_{k'}(D) + L_{k'}\right) | (L_{k'})_{k' \neq k}\right) . \tag{49}$$

Notice that we took strict inequalities ($<$) because $L_k$ is absolutely continuous w.r.t. Lebesgue's measure conditionally on $(L_{k'})_{k' \neq k}$, and hence, ties occur only with null probability.

Thus,

$$\begin{aligned}
\mathbb{P}\left(\hat{i}(D) = k | (L_{k'})_{k' \neq k}\right) &= \mathbb{P}\left(L_k < \min_{k' \neq k} \left(f_{k'}(D) - f_k(D) + L_{k'}\right) \Big| (L_{k'})_{k' \neq k}\right) \\
&= \frac{1}{2} \int_{-\infty}^{\min_{k' \neq k} (f_{k'}(D) - f_k(D) + L_{k'})} e^{-|t|} dt .
\end{aligned} \tag{50}$$

Hence, two cases occur

**Case A:** $\min_{k' \neq k} \left(f_{k'}(D) - f_k(D) + L_{k'}\right) \leq 0$

$$\mathbb{P}\left(\hat{i}(D) = k | (L_{k'})_{k' \neq k}\right) = \frac{1}{2} e^{\min_{k' \neq k} (f_{k'}(D) - f_k(D) + L_{k'})} . \tag{51}$$

**Case B:** $\min_{k' \neq k} \left(f_{k'}(D) - f_k(D) + L_{k'}\right) > 0$

$$\mathbb{P}\left(\hat{i}(D) = k | (L_{k'})_{k' \neq k}\right) = 1 - \frac{1}{2} e^{-\min_{k' \neq k} (f_{k'}(D) - f_k(D) + L_{k'})} . \tag{52}$$

Now, we can look at the ratio $\frac{\mathbb{P}(\hat{i}(D)=k|(L_{k'})_{k'\neq k})}{\mathbb{P}(\hat{i}(D')=k|(L_{k'})_{k'\neq k})}$. First, we can notice that

$$\left| \left( \min_{k'\neq k} (f_{k'}(D) - f_k(D) + L_{k'}) \right) - \left( \min_{k'\neq k} (f_{k'}(D') - f_k(D') + L_{k'}) \right) \right| \leq 2\Delta . \tag{53}$$

Indeed, since the sensitivities of $(f_k)_k$ are uniformly bounded, for any $k' \neq k$,

$$f_{k'}(D) - f_k(D) \leq (f_{k'}(D') + \Delta) - (f_k(D') - \Delta) = f_{k'}(D') - f_k(D') + 2\Delta , \tag{54}$$

and taking the minimum yields half of the result. The other half is given by swapping $D$ and $D'$.

Then, by using $t(D)$ as a short for $\min_{k'\neq k}(f_{k'}(D) - f_k(D) + L_{k'})$, we can look at the following four cases:

**Case 1:** $t(D), t(D') \leq 0$

$$\frac{\mathbb{P}\left(\hat{i}(D) = k|(L_{k'})_{k'\neq k}\right)}{\mathbb{P}\left(\hat{i}(D') = k|(L_{k'})_{k'\neq k}\right)} \overset{(51)}{=} \frac{\frac{1}{2}e^{t(D)}}{\frac{1}{2}e^{t(D')}} \leq e^{|t(D)-t(D')|} \overset{(53)}{\leq} e^{2\Delta} . \tag{55}$$

**Case 2:** $t(D) \leq 0 < t(D')$

$$\frac{\mathbb{P}\left(\hat{i}(D) = k|(L_{k'})_{k'\neq k}\right)}{\mathbb{P}\left(\hat{i}(D') = k|(L_{k'})_{k'\neq k}\right)} \overset{(51)\&(52)}{=} \frac{\frac{1}{2}e^{t(D)}}{1 - \frac{1}{2}e^{-t(D')}} = e^{t(D')-t(D)} \frac{\frac{1}{2}e^{t(D)-t(D')}}{e^{-t(D)} - \frac{1}{2}e^{-t(D')-t(D)}} . \tag{56}$$

Since $t(D) \leq 0 < t(D')$,

$$e^{t(D)-t(D')} \leq 1 . \tag{57}$$

Furthermore, a simple study of the function $x \mapsto e^{-x} - \frac{1}{2}e^{-t(D')-x}$ on $(-\infty, 0]$ shows that it is non-increasing (because $t(D') > 0$), and thus that

$$e^{-t(D)} - \frac{1}{2}e^{-t(D')-t(D)} \geq 1 - \frac{1}{2}e^{-t(D')} \geq \frac{1}{2} . \tag{58}$$

In the end,

$$\frac{\mathbb{P}\left(\hat{i}(D) = k|(L_{k'})_{k'\neq k}\right)}{\mathbb{P}\left(\hat{i}(D') = k|(L_{k'})_{k'\neq k}\right)} \leq e^{t(D')-t(D)} \overset{(53)}{\leq} e^{2\Delta} . \tag{59}$$

**Case 3:** $t(D') \leq 0 < t(D)$

$$\frac{\mathbb{P}\left(\hat{i}(D) = k|(L_{k'})_{k'\neq k}\right)}{\mathbb{P}\left(\hat{i}(D') = k|(L_{k'})_{k'\neq k}\right)} \overset{(51)\&(52)}{=} \frac{1 - \frac{1}{2}e^{-t(D)}}{\frac{1}{2}e^{t(D')}} = e^{t(D)-t(D')} \frac{e^{-t(D)} - \frac{1}{2}e^{-t(D')-t(D)}}{\frac{1}{2}}$$

$$\leq e^{t(D)-t(D')} \frac{e^{-t(D)} - \frac{1}{2}e^{-t(D)}}{\frac{1}{2}} = e^{t(D)-t(D')} \frac{\frac{1}{2}e^{-t(D)}}{\frac{1}{2}} \tag{60}$$

$$\leq e^{t(D)-t(D')} \overset{(53)}{\leq} e^{2\Delta} .$$

**Case 4:** $t(D), t(D') > 0$

$$\frac{\mathbb{P}\left(\hat{i}(D) = k | (L_{k'})_{k' \neq k}\right)}{\mathbb{P}\left(\hat{i}(D') = k | (L_{k'})_{k' \neq k}\right)} \overset{(52)}{=} \frac{1 - \frac{1}{2}e^{-t(D)}}{1 - \frac{1}{2}e^{-t(D')}}$$

$$\leq \min\left\{ e^{t(D')-t(D)} \underbrace{\frac{e^{-t(D')} - \frac{1}{2}e^{-t(D)-t(D')}}{e^{-t(D)} - \frac{1}{2}e^{-t(D')-t(D)}}}_{\leq 1 \text{ if } t(D') \geq t(D)}, e^{t(D)-t(D')} \underbrace{\frac{e^{-t(D)} - \frac{1}{2}e^{-2t(D)}}{e^{-t(D')} - \frac{1}{2}e^{-2t(D')}}}_{\leq 1 \text{ if } t(D) \geq t(D')} \right\} \quad (61)$$

$$\leq e^{|t(D)-t(D')|}$$

$$\overset{(53)}{\leq} e^{2\Delta}.$$

Thus, we have shown that $\frac{\mathbb{P}\left(\hat{i}(D)=k|(L_{k'})_{k' \neq k}\right)}{\mathbb{P}\left(\hat{i}(D')=k|(L_{k'})_{k' \neq k}\right)} \leq e^{2\Delta}$.

Integrating over $(L_{k'})_{k' \neq k}$ yields

$$\mathbb{E}_{(L_{k'})_{k' \neq k}}\left( \mathbb{P}\left(\hat{i}(D) = k | (L_{k'})_{k' \neq k}\right)\right) \leq e^{2\Delta} \mathbb{E}_{(L_{k'})_{k' \neq k}}\left( \mathbb{P}\left(\hat{i}(D') = k | (L_{k'})_{k' \neq k}\right)\right), \quad (62)$$

which translates to

$$\mathbb{P}\left(\hat{i}(D) = k\right) \leq e^{2\Delta} \mathbb{P}\left(\hat{i}(D') = k\right). \quad (63)$$

Finally, since this is true for any $k$, $\hat{i}$ is $\epsilon$-DP when $\Delta \leq \frac{\epsilon}{2}$. This concludes the proof by scaling appropriatetly.

### D.2. Proof of Lemma 3.3

Let $(X_{1:n}, Y_{1:n}) \sim (X'_{1:n}, Y'_{1:n})$ be two neighboring datasets. By definition, they contain the same data points except for one individual. Suppose first that $Y_{1:n} = Y'_{1:n}$ and that, for some $i \in [n]$, the datasets $X_{1:n}$ and $X'_{1:n}$ differ in the $i$-th coordinate only. Then the term $\frac{2\|f\|_\infty}{n}$ is obtained by noting that, by the triangular inequality

$$\left| \hat{S}(f|(X_{1:n}, Y_{1:n})) - \hat{S}(f|(X'_{1:n}, Y_{1:n})) \right| = \frac{1}{n}|f(X_i) - f(X'_i)| \leq \frac{2\|f\|_\infty}{n}.$$

In the opposite case, it holds that $X_{1:n} = X'_{1:n}$ and that the datasets $Y_{1:n}$ and $Y'_{1:n}$ differ in one of the coordinates $i \in [n]$. We first prove that $f^*$ is $|\tilde{\Omega}|$-Lipschitz on $\tilde{\Omega}$.

Let $y_1, y_2 \in \tilde{\Omega}$, let $x \in \tilde{\Omega}$,

$$\langle x, y_1 \rangle - f(x) = \langle x, y_1 - y_2 \rangle + \langle x, y_2 \rangle - f(x)$$

$$\overset{\text{Cauchy-Schwarz}}{\leq} \|x\|\|y_1 - y_2\| + \langle x, y_2 \rangle - f(x) \quad (64)$$

$$\leq |\tilde{\Omega}|\|y_1 - y_2\| + \langle x, y_2 \rangle - f(x).$$

Thus, taking the sup with respect to $x$ on the left-hand side and on the right-hand side yields

$$f^*(y_1) \leq |\tilde{\Omega}|\|y_1 - y_2\| + f^*(y_2), \quad (65)$$

and since $y_1$ and $y_2$ play symmetric roles,

$$f^*(y_2) \leq |\tilde{\Omega}|\|y_1 - y_2\| + f^*(y_1). \quad (66)$$

Hence, $\hat{S}(f|X_{1:n}, Y_{1:n})$ is $\frac{|\tilde{\Omega}|}{n}$-Lipschitz w.r.t. any of the $Y$s, and therefore, changing the value of any of the $Y$s can only affect the value of $\hat{S}(f|X_{1:n}, Y_{1:n})$ by at most $\frac{2|\tilde{\Omega}|^2}{n}$ by a triangular inequality.

# E. Proofs of Section 4

### E.1. Proof of Lemma 4.1

Let $f_1, f_2 \in \mathcal{X}$. We observe that $\|f_1^* - f_2^*\|_\infty \leq \|f_1 - f_2\|_\infty$, since for any $y \in \tilde{\Omega}$,

$$
\begin{aligned}
f_1^*(y) - f_2^*(y) &=: \sup_x \left\{ \langle x, y \rangle - f_1(x) \right\} + \inf_{x'} \left\{ -\langle x', y \rangle + f_2(x') \right\} \\
&= \sup_x \inf_{x'} \langle x - x', y \rangle + f_2(x') - f_1(x) \\
&\leq \sup_x \inf_{x'=x} \langle x - x', y \rangle + f_2(x') - f_1(x) \\
&= \sup_x \ f_2(x) - f_1(x).
\end{aligned}
$$

Thus,

$$
\begin{aligned}
\left| \hat{S}(f_1|X_{1:n}, Y_{1:n}) - \hat{S}(f_2|X_{1:n}, Y_{1:n}) \right| &= \left| \frac{1}{n} \sum_{i=1}^n (f_1(X_i) - f_2(X_i)) + \frac{1}{n} \sum_{i=1}^n (f_1^*(Y_i) - f_2^*(Y_i)) \right| \\
&\leq \|f_1 - f_2\|_\infty + \|f_1^* - f_2^*\|_\infty \\
&\leq 2\|f_1 - f_2\|_\infty .
\end{aligned} \tag{67}
$$

### E.2. Proof of Lemma 4.2

By Lemma B.3, we have

$$
V_J \cap \mathcal{X}(2M) \subset B_{V_J, \|\cdot\|_{\infty,J}}(0, CM^2) \tag{68}
$$

for a positive constant $C$, where $B_{V_J, \|\cdot\|_{\infty,J}}(0, r)$ refers to the closed ball of $V_J$ with center 0 and radius $r \geq 0$ for the norm $\| \cdot \|_{\infty,J}$ (the infinite norm of the vector of coefficients in the wavelet basis).

Since this is a ball in a finite-dimensional space for a distance in infinite norm, it is possible to build a $\delta$-covering of this ball with a $\delta$-grid on the coefficients of the wavelet decomposition, which is of cardinality

$$
C_J := \left( \left\lceil \frac{2CM^2}{\delta} \right\rceil \right)^{\dim(V_J)} . \tag{69}
$$

Furthermore, because of the bounded support of the father and mother wavelets, $\dim(V_J) \lesssim 2^{Jd}$ (see page 30 in (Hütter & Rigollet, 2021)). Thus,

$$
C_J \leq \left( \left\lceil \frac{2CM^2}{\delta} \right\rceil \right)^{C'2^{Jd}} \tag{70}
$$

where $C'$ is a constant that does not depend on $J$ and $\delta$.

Let us consider the following construction: for each element of this covering, if there exists a point in $V_J \cap \mathcal{X}(2M)$ at distance not more than $\delta$, we replace the original element by this new point (we choose any point that satisfies this condition in case of non uniqueness). Else, we remove this element if no point in $V_J \cap \mathcal{X}(2M)$ satisfies this condition. By the triangular inequality, this construction yields a $2\delta$ covering of $V_J \cap \mathcal{X}(2M)$ for $\| \cdot \|_{\infty,J}$ of cardinality at most $C_J$.

Furthermore, by Lemma B.3, a $\frac{\delta}{2^{Jd/2}}$-covering of $V_J \cap \mathcal{X}(2M)$ for $\| \cdot \|_{\infty,J}$ is a $\delta$-covering of $V_J \cap \mathcal{X}(2M)$ for the usual infinite norm $\| \cdot \|_\infty$.

Scaling $\delta$ appropriately concludes the proof.

### E.3. Proof of Theorem 4.3

We will control

$$
\hat{S}(\hat{f}_{\text{priv}}|X_{1:n}, Y_{1:n}) - \hat{S}(\hat{f}_J|X_{1:n}, Y_{1:n}) . \tag{71}
$$

We work conditionally on $X_{1:n}, Y_{1:n}$. All the relations below hold almost surely.

Since $C_{J,M}$ is a $\delta$-covering of $V_J \cap \mathcal{X}(2M)$ with respect to $\|\cdot\|_\infty$, it is also a $2\delta$-covering of $V_J \cap \mathcal{X}(2M)$ with respect to $\hat{S}(\cdot|X_{1:n}, Y_{1:n})$ by Lemma 4.1. Thus,

$$\hat{S}(\hat{f}_{\text{priv}}|X_{1:n}, Y_{1:n}) - \hat{S}(\hat{f}_J|X_{1:n}, Y_{1:n})$$

$$= \min_{i \in \{1,\dots,\#(C_{J,M})\}} \left\{ \hat{S}(f_i|D) + \frac{32M^2 \vee 36d}{n\epsilon} L_i \right\} - \hat{S}(\hat{f}_J|X_{1:n}, Y_{1:n})$$

$$\leq \min_{i \in \{1,\dots,\#(C_{J,M})\}} \left\{ \hat{S}(f_i|X_{1:n}, Y_{1:n}) - \hat{S}(\hat{f}_J|X_{1:n}, Y_{1:n}) \right\} + \max_i \left| \frac{32M^2 \vee 36d}{n\epsilon} L_i \right| \qquad (72)$$

$$\leq 2 \min_{i \in \{1,\dots,\#(C_{J,M})\}} \|f_i - \hat{f}_J\|_\infty + \max_i \left| \frac{32M^2 \vee 36d}{n\epsilon} L_i \right|$$

$$\leq \underbrace{2\delta + \max_i \left| \frac{32M^2 \vee 36d}{n\epsilon} L_i \right|}_{=:D} .$$

By integrating over the source of privacy and by Lemma B.2,

$$\mathbb{E}_{L_{1:\#(C_{J,M})}}(D) \lesssim \delta + \frac{\ln(\#(C_{J,M}))}{n\epsilon}$$

$$\lesssim \delta + \frac{2^{Jd} \ln\left(\frac{C2^{Jd/2}}{\delta} + 1\right)}{n\epsilon} , \qquad (73)$$

where $C \geq 0$ is a constant independent of $J$ and $\delta$. Fixing $\delta = \frac{C2^{Jd/2}}{n\epsilon}$ yields

$$\mathbb{E}_{L_{1:\#(C_{J,M})}}(D) \lesssim \frac{2^{Jd} \ln(n\epsilon)}{n\epsilon} . \qquad (74)$$

By Lemma 2.5 and subsection 5.5 in (Hütter & Rigollet, 2021) in order to control the bias,

$$\mathbb{E}\left(d(\hat{T}_{\text{priv}}, T_0)^2\right) \lesssim \frac{2^{Jd} \ln(n\epsilon)}{n\epsilon} + \frac{J2^{J(d-2)} \ln(n)}{n} + \frac{1}{n} + R^2 2^{-2J\alpha} \qquad (75)$$

for $J$ larger than a suitable constant.

# F. Proofs of Section 5

## F.1. Proof of Theorem 5.1

The proof of the lower bounding terms $\frac{1}{n}$ and $n^{-\frac{2\alpha}{2\alpha-2+d}}$ uses standard packing arguments (Tsybakov, 2009) with a specific packing for the problem at hand. It may be found in (Hütter & Rigollet, 2021). We only provide the proof for the additional term $(n\epsilon)^{-\frac{2\alpha}{2\alpha+d}}$. It is based on packing arguments for differential privacy standardized via couplings in (Acharya et al., 2018; 2021; Lalanne et al., 2023a). We start by building a packing similar to the one from (Hütter & Rigollet, 2021), which is later analyzed with TV distances instead of the traditional KL divergences in order to highlight the effect of differential privacy.

**Packing construction.** Let $m$ be an integer that will be specified later in the proof and let $N = m^d$. We consider the grid $\left(\left(\frac{k_1}{m+1}, \ldots, \frac{k_d}{m+1}\right)\right)_{1 \leq k_1, \ldots, k_d \leq m}$. This grid has cardinality $N$, therefore we can enumerate its elements as $(p_1, \ldots, p_N) \in (\mathbb{R}^d)^N$. By construction, we have that

$$\forall i, j \in \{1, \ldots, N\}, \quad i \neq j \implies \|p_i - p_j\|_\infty \geq \frac{1}{m+1} . \tag{76}$$

Now, we consider a $C^\infty$ "bump" function $B$ defined over $\mathbb{R}$ and supported on $[-1, 1]$, taking positive values over $(-1, 1)$. For such a function, there exists $x_0 \in (-1, 1)$ such that $B(x_0) \neq 0$ and $B'(x_0) \neq 0$.

For some sufficiently small constant $a > 0$ whose value will be fixed later, we define the function

$$\psi(x_1, \ldots, x_d) = a \prod_{i=1}^{d} B\left(\frac{x_i}{2}\right) . \tag{77}$$

and for some $h \in (0, 1]$ and any $\theta \in \{0, 1\}^N$, we define the potential $\phi_\theta : \tilde{\Omega} \to \mathbb{R}$ as

$$\phi_\theta(\cdot) := \frac{1}{2}\|\cdot\|^2 + h^{\alpha+1} \sum_{i=1}^{N} \theta_i \psi\left(\frac{\cdot - p_i}{h}\right) . \tag{78}$$

To ensure that the functions $\psi\left(\frac{\cdot - p_i}{h}\right)$ for $i \in \{1, \ldots, N\}$ have disjoint supports, we take $h < \frac{1}{2(m+1)} \lesssim \frac{1}{m}$.

First, we show that for any $\theta \in \{0, 1\}^N$, $\nabla\phi_\theta \in \mathcal{T}_\alpha$:

- (Bounded transport map) $\forall \theta \in \{0, 1\}^N, \forall x \in \tilde{\Omega}$,

$$\nabla\phi_\theta(x) = x + h^\alpha \sum_{i=1}^{N} \theta_i \nabla\psi\left(\frac{x - p_i}{h}\right) , \tag{79}$$

  and since the supports are disjoint, only one term in the sum is non-zero, which yields

$$\forall \theta \in \{0, 1\}^N, \forall x \in \tilde{\Omega}, \quad \|\nabla\phi_\theta(x) - x\| \leq h^\alpha \|\nabla\psi\|_\infty. \tag{80}$$

  Choosing $a$ small enough ensures that $\forall \theta \in \{0, 1\}^N, \forall x \in \tilde{\Omega}, \|T(x)\| \leq M$.

- (Strong convexity of the potential) $\forall \theta \in \{0, 1\}^N, \forall x \in \tilde{\Omega}$,

$$\nabla^2\phi_\theta(x) = I_d + h^{\alpha-1} \sum_{i=1}^{N} \theta_i \nabla^2\psi\left(\frac{x - p_i}{h}\right) \tag{81}$$

  and thus, since the supports are disjoint,

$$\forall \theta \in \{0, 1\}^N, \forall x \in \tilde{\Omega}, \quad \|\nabla^2\phi_\theta(x) - I_d\|_{\text{op}} \leq h^{\alpha-1} \sup \|\nabla^2\psi\|_{\text{op}} , \tag{82}$$

  where the right-hand side can be made arbitrarily close to 0 by taking $a$ small enough (since $h^{\alpha-1} \leq 1$), which ensures that $\forall \theta \in \{0, 1\}^N, \forall x \in \tilde{\Omega}, \frac{1}{M} \preceq \nabla^2\phi_\theta(x) \preceq M$.

- (Stability of the support) Let $\theta \in \{0,1\}^N$. It follows from (80) that, when $h \leq \frac{1}{m+1}$, one can take $a$ small enough to ensure that

$$\forall i, \nabla\phi_\theta \left( B_\infty \left( p_i, \frac{h}{2} \right) \right) \subset B_\infty \left( p_i, h \right) \ . \tag{83}$$

Furthermore, since outside the balls of the form $\left( B_\infty \left( p_i, \frac{h}{2} \right) \right)$, $\nabla\phi_\theta$ is the identity, we have that $\nabla\phi_\theta(\Omega) \subset \Omega$ and that

$$\forall i, \nabla\phi_\theta \left( B_\infty \left( p_i, h \right) \right) \subset B_\infty \left( p_i, h \right) \ . \tag{84}$$

We now give additional details on the properties of $\nabla\phi_\theta$. We have proved above that $\phi_\theta$ is strongly convex for any $\theta \in \{0,1\}^N$. It follows that $\nabla\phi_\theta$ is injective and that, for any $x \in \tilde{\Omega}$, the Hessian $\nabla^2\phi_\theta(x) \succ 0$ is invertible. Since $\nabla\phi_\theta$ is $\mathcal{C}^\infty$ over $\tilde{\Omega}$, the global inversion theorem states that $\nabla\phi_\theta$ is a $\mathcal{C}^\infty$ diffeomorphism from $\tilde{\Omega}$ to $\nabla\phi_\theta(\tilde{\Omega})$.

We may also see that the inclusion $\nabla\phi_\theta(\Omega) \subset \Omega$ is in fact an equality using arguments borrowed from basic algebraic topology. First, $\nabla\phi_\theta$ is the identity over $\partial\Omega$. Suppose for the sake of contradiction that $\Omega \setminus \nabla\phi_\theta(\Omega) \neq \emptyset$ and let $x \in \Omega \setminus \nabla\phi_\theta(\Omega)$ ($x \notin \partial\Omega$ since $\nabla\phi_\theta$ is the identity over $\partial\Omega$). Then, letting $r_x$ be a retraction of $\Omega \setminus \{x\}$ on $\partial\Omega$ (which exists), we obtain that $r_x \circ \nabla\phi_\theta$ is a retraction of $\Omega$ (which is homeomorphic to $B(0,1)$) to $\partial\Omega$ (which is homeomorphic to $\partial B(0,1)$), which is impossible by Brouwer's theorem (see Theorem 7.6.2 in (Randal-Williams, 2024)).

As a consequence,

$$\forall \theta \in \{0,1\}^N, \quad \nabla\phi_\theta(\Omega) = \Omega \ . \tag{85}$$

By injectivity of the functions $\phi_\theta$'s, this immediately yields

$$\forall \theta \in \{0,1\}^N, \quad (\nabla\phi_\theta)^{-1}(\Omega) = \Omega \ . \tag{86}$$

Finally, the same arguments applied to the supports of the functions $\psi\left(\frac{\cdot - p_i}{h}\right)$ ensure that, for a small enough $a > 0$, we have

$$\forall \theta \in \{0,1\}^N, \forall i, \quad \nabla\phi_\theta \left( B_\infty \left( p_i, h \right) \right) = B_\infty \left( p_i, h \right) \ . \tag{87}$$

By injectivity of the functions $\phi_\theta$'s, we similarly obtain

$$\forall \theta \in \{0,1\}^N, \forall i, \quad (\nabla\phi_\theta)^{-1} \left( B_\infty \left( p_i, h \right) \right) = B_\infty \left( p_i, h \right) \ . \tag{88}$$

- (Smoothness) Let $\theta \in \{0,1\}^N$. Using multi-index notation (see (Hütter & Rigollet, 2021)),

$$\|\nabla\phi_\theta\|_{C^\alpha(\tilde{\Omega})} = \sum_{i=1}^d \left( \sum_{|b| \leq \alpha} \underbrace{\|\partial^b (\nabla\phi_\theta)_i\|_\infty}_{=:A_{i,b}} + \sum_{|b| = \lfloor\alpha\rfloor} \underbrace{\sup_{x \neq y, x, y \in \tilde{\Omega}} \frac{|\partial^b (\nabla\phi_\theta)_i(x) - \partial^b (\nabla\phi_\theta)_i(y)|}{|x-y|^{\alpha - \lfloor\alpha\rfloor}}}_{=:B_{i,b}} \right) \ . \tag{89}$$

Furthermore,

$$A_{i,b} = \left\| \partial^b \text{Coord.}_i + h^{\alpha-|b|} \sum_{j=1}^N \theta_j \partial^b (\nabla\psi)_i \left( \frac{\cdot - p_j}{h} \right) \right\|_\infty \tag{90}$$

where $\text{Coord.}_i$ is the projection along the $i^{\text{th}}$ canonical vector. Thus, $A_{i,b}$ can be made as close as we want to $\left\| \partial^b \text{Coord.}_i \right\|_\infty$ granted that the constant $a$ is taken small enough.

Finally,

$$B_{i,b} = \sup_{x \neq y, x, y \in \tilde{\Omega}}$$

$$\frac{\left| \partial^b \operatorname{Coord.}_i(x) + h^{\alpha - |b|} \sum_{j=1}^N \theta_j \partial^b (\nabla \psi)_i \left( \frac{x - p_j}{h} \right) - \partial^b \operatorname{Coord.}_i(y) + h^{\alpha - |b|} \sum_{j=1}^N \theta_j \partial^b (\nabla \psi)_i \left( \frac{y - p_j}{h} \right) \right|}{|x - y|^{\alpha - \lfloor \alpha \rfloor}}$$

$$= \sup_{x \neq y, x, y \in \tilde{\Omega}} \frac{\left| h^{\alpha - |b|} \sum_{j=1}^N \theta_j \partial^b (\nabla \psi)_i \left( \frac{x - p_j}{h} \right) - h^{\alpha - |b|} \sum_{j=1}^N \theta_j \partial^b (\nabla \psi)_i \left( \frac{y - p_j}{h} \right) \right|}{|x - y|^{\alpha - \lfloor \alpha \rfloor}}$$

$$\leq h^{\alpha - |b|} \sup_{x \neq y, x, y \in \tilde{\Omega}} \frac{\sum_{j=1}^N \left| \partial^b (\nabla \psi)_i \left( \frac{x - p_j}{h} \right) - \partial^b (\nabla \psi)_i \left( \frac{y - p_j}{h} \right) \right|}{|x - y|^{\alpha - \lfloor \alpha \rfloor}} \tag{91}$$

$$\overset{\text{Disjoint Supports \& Continuity}}{\leq} h^{\alpha - |b|} \max_i \sup_{x \neq y, x, y \in \tilde{\Omega}} \frac{\left| \partial^b (\nabla \psi)_i \left( \frac{x - p_j}{h} \right) - \partial^b (\nabla \psi)_i \left( \frac{y - p_j}{h} \right) \right|}{|x - y|^{\alpha - \lfloor \alpha \rfloor}}$$

$$\leq h^{\alpha - |b|} \max_i \sup_{x \neq y, x, y \in \mathbb{R}^d} \frac{\left| \partial^b (\nabla \psi)_i \left( \frac{x - p_j}{h} \right) - \partial^b (\nabla \psi)_i \left( \frac{y - p_j}{h} \right) \right|}{|x - y|^{\alpha - \lfloor \alpha \rfloor}}$$

$$= h^{\alpha - |b|} \max_i h^{-(\alpha - \lfloor \alpha \rfloor)} \sup_{x \neq y, x, y \in \mathbb{R}^d} \frac{\left| \partial^b (\nabla \psi)_i (x) - \partial^b (\nabla \psi)_i (y) \right|}{|x - y|^{\alpha - \lfloor \alpha \rfloor}}$$

$$= \max_i \sup_{x \neq y, x, y \in \mathbb{R}^d} \frac{\left| \partial^b (\nabla \psi)_i (x) - \partial^b (\nabla \psi)_i (y) \right|}{|x - y|^{\alpha - \lfloor \alpha \rfloor}} .$$

Hence, $B_{i,b}$ can be made as small as we want to 0 granted that the constant $a$ is taken small enough.

All in all, if the constant $a$ is taken small enough, $\sup_\theta \|\nabla \phi_\theta\|_{C^\alpha(\tilde{\Omega})}$ can be made as close as we want to $\|\operatorname{Identity}\|_{C^\alpha(\tilde{\Omega})}$, which guarantees that $\sup_\theta \|\nabla \phi_\theta\|_{C^\alpha(\tilde{\Omega})} \leq R$ if $a$ is small enough.

Similarly as in (Hütter & Rigollet, 2021), we consider the family of statistics $(S_\theta := P^{\otimes n} \otimes Q_\theta^{\otimes n})_{\theta \in \{0,1\}^N}$ on the problem where

$$P = \operatorname{Unif}([0,1]^d) , \tag{92}$$

and

$$\forall \theta \in \{0,1\}^N, \quad Q_\theta = P_{\#\nabla \phi_\theta} . \tag{93}$$

It is thus possible to simplify the lower-bounding problem as

$$\inf_{\hat{T}} \sup_{P \in \mathcal{M}, T_0 \in \mathcal{T}_\alpha} \mathbb{E} \left( \int \|T_0 - \hat{T}\|^2 dP \right) \geq \inf_{\hat{T}} \sup_{\theta \in \{0,1\}^N} \mathbb{E}_{S_\theta} \left( \int \|\nabla \phi_\theta - \hat{T}\|^2 dP \right) \tag{94}$$

where the supremum is now taken over a finite family only.

We verify that this family of statistics forms a packing for the pseudo metric defined below (with some overlap in the notation)

$$d(S_{\theta_1}, S_{\theta_2})^2 := d(\theta_1, \theta_2)^2 := \int \|\nabla \phi_{\theta_1} - \nabla \phi_{\theta_2}\|^2 dP \tag{95}$$

**Lemma F.1.** *For any $\theta_1, \theta_2 \in \{0,1\}^N$,*

$$d(S_{\theta_1}, S_{\theta_2})^2 \gtrsim d_{\operatorname{ham}}(\theta_1, \theta_2) h^{2\alpha + d} . \tag{96}$$

*Proof.* Let $\theta_1, \theta_2 \in \{0,1\}^N$, then

$$
\int_{[0,1]^d} \|\nabla\phi_{\theta_1} - \nabla\phi_{\theta_2}\|^2 \overset{\text{Disjoint supports}}{=} \sum_{i=1}^{N} \int_{[0,1]^d} \left\| (\theta_1^i - \theta_2^i) h^\alpha \nabla\psi\left(\frac{x - p_i}{h}\right) \right\|^2 dx
$$

$$
\overset{\text{Change of variables}}{=} d_{\text{ham}}(\theta_1, \theta_2) h^{2\alpha+d} \underbrace{\int \|\nabla\psi\|^2}_{>0} \tag{97}
$$

$$
\gtrsim d_{\text{ham}}(\theta_1, \theta_2) h^{2\alpha+d} .
$$

$\square$

From line 1 to line 2, we used that $|\theta_1^i - \theta_2^i|$ is equal to 1 only $d_{\text{ham}}(\theta_1, \theta_2)$ times, and is it equal to 0 in the other cases. The change of variable $u = \frac{x-p_i}{h}$ in each of the remaining terms yields the term $h^d$.

**Upper bounding the TV distances.** To proceed with the lower-bound argument, it remains to lower-bound the testing difficulty between the hypotheses of this packing. Techniques for bounding the testing difficulties usually require bounding the KL-divergences between the statistical models of the packing and a reference measure. This is the approach proposed in (Hütter & Rigollet, 2021), which leads to a lower bound of the order of $\frac{1}{n} \vee n^{-\frac{2\alpha}{2\alpha-2+d}}$. However, under differential privacy, the test difficulty is characterized by the TV distances between the *untensorized* marginals of the statistical models. This part of the proof is dedicated to controlling these terms.

**Lemma F.2.** *For any $\theta_1, \theta_2 \in \{0,1\}^N$, it holds that*

$$
\text{TV}(Q_{\theta_1}, Q_{\theta_2}) \lesssim d_{\text{ham}}(\theta_1, \theta_2) h^{\alpha-1+d} . \tag{98}
$$

*Proof.* By the change of variables formula, for any $\theta \in \{0,1\}^N$, the density of $Q_\theta$ with respect to $P$ is

$$
\frac{dQ_\theta}{dP}(y) = \frac{\mathbb{1}_{[0,1]^d}((\nabla\phi_\theta)^{-1}(y))}{\det(\nabla^2\phi_\theta((\nabla\phi_\theta)^{-1}(y)))} \tag{99}
$$

for $P$-almost all $y$. Let $\theta_1, \theta_2 \in \{0,1\}^N$,

$$
\text{TV}(Q_{\theta_1}, Q_{\theta_2}) = \frac{1}{2} \int_{(-1,2)^d} \left| \frac{dQ_{\theta_1}}{dP}(y) - \frac{dQ_{\theta_2}}{dP}(y) \right| dP(y)
$$

$$
= \frac{1}{2} \int_{[0,1]^d} \left| \frac{\mathbb{1}_{[0,1]^d}((\nabla\phi_{\theta_1})^{-1}(y))}{\det(\nabla^2\phi_{\theta_1}((\nabla\phi_{\theta_1})^{-1}(y)))} - \frac{\mathbb{1}_{[0,1]^d}((\nabla\phi_{\theta_2})^{-1}(y))}{\det(\nabla^2\phi_{\theta_2}((\nabla\phi_{\theta_2})^{-1}(y)))} \right| dy , \tag{100}
$$

and by stability of $[0,1]^d$ and of its complement by either $\nabla\phi_{\theta_1}$ or $\nabla\phi_{\theta_2}$, this expression can be further simplified as

$$
\text{TV}(Q_{\theta_1}, Q_{\theta_2}) = \frac{1}{2} \int_{[0,1]^d} \left| \frac{1}{\det(\nabla^2\phi_{\theta_1}((\nabla\phi_{\theta_1})^{-1}(y)))} - \frac{1}{\det(\nabla^2\phi_{\theta_2}((\nabla\phi_{\theta_2})^{-1}(y)))} \right| dy . \tag{101}
$$

Then, since both $\nabla\phi_{\theta_1}$ and $\nabla\phi_{\theta_2}$ are the identity outside balls of the form $B_\infty(p_i, h)$, and since these balls and their complements are stable by $\nabla\phi_{\theta_1}$ and by $\nabla\phi_{\theta_2}$, this can be rewritten as

$$
\text{TV}(Q_{\theta_1}, Q_{\theta_2}) = \frac{1}{2} \sum_{i=1}^{N} \int_{B_\infty(p_i, h)} \left| \frac{1}{\det(\nabla^2\phi_{\theta_1}((\nabla\phi_{\theta_1})^{-1}(y)))} - \frac{1}{\det(\nabla^2\phi_{\theta_2}((\nabla\phi_{\theta_2})^{-1}(y)))} \right| dy . \tag{102}
$$

This expression finally further simplifies as

$$
\text{TV}(Q_{\theta_1}, Q_{\theta_2}) =
$$

$$
\frac{1}{2} \sum_{i=1}^{N} \mathbb{1}_{\theta_1^{(i)} \neq \theta_2^{(i)}} \underbrace{\int_{B_\infty(p_i, h)} \left| \frac{1}{\det(\nabla^2\phi_{\theta_1}((\nabla\phi_{\theta_1})^{-1}(y)))} - \frac{1}{\det(\nabla^2\phi_{\theta_2}((\nabla\phi_{\theta_2})^{-1}(y)))} \right| dy}_{:=\Delta_i} . \tag{103}
$$

We treat the case $\theta_1^{(i)} = 0$, $\theta_2^{(i)} = 1$, the other case being treated similarly. In this case,

$$
\begin{aligned}
\Delta_i &= \int_{B_\infty(p_i,h)} \left| 1 - \frac{1}{\det(\nabla^2\phi_{\theta_2}((\nabla\phi_{\theta_2})^{-1}(y)))} \right| dy \\
&= \int_{B_\infty(p_i,h)} \left| 1 - \frac{1}{\det(\nabla^2\phi_{\theta_2}(x))} \right| |\det(\nabla^2\phi_{\theta_2}(x))| dx \ ,
\end{aligned}
\tag{104}
$$

where the last line comes from a change of variables and from the stability of $B_\infty(p_i,h)$ and of its complement.

For any $x \in B_\infty(p_i,h)$,

$$
\nabla^2\phi_{\theta_2}(x) = I_d + h^{\alpha-1}\nabla^2\psi\left(\frac{x-p_i}{h}\right) \ .
\tag{105}
$$

By Lemma B.1, we see that if $a$ is chosen small enough, then we have $|\det(\nabla^2\phi_{\theta_2}(\cdot))| \leq 2$ uniformly over $B_\infty(p_i,h)$. Finally, and again by Lemma B.1, if $a$ is chosen small enough, $\forall x \in B_\infty(p_i,h)$,

$$
\left| 1 - \frac{1}{\det(\nabla^2\phi_{\theta_2}(x))} \right| \leq h^{\alpha-1}\left( \left| \mathrm{tr}\left(\nabla^2\psi\left(\frac{x-p_i}{h}\right)\right) \right| + \left\| \nabla^2\psi\left(\frac{x-p_i}{h}\right) \right\| \right) \ ,
\tag{106}
$$

which entails, by a change of variables,

$$
\Delta_i \leq \frac{h^{\alpha-1+d}}{2}\int \left( \left| \mathrm{tr}\left(\nabla^2\psi\right) \right| + \left\| \nabla^2\psi \right\| \right) \lesssim h^{\alpha-1+d} \ .
\tag{107}
$$

Plugging this result back into (103) yields

$$
\begin{aligned}
\mathrm{TV}\left(Q_{\theta_1}, Q_{\theta_2}\right) &\lesssim h^{\alpha-1+d}\sum_{i=1}^N \mathbb{1}_{\theta_1^{(i)} \neq \theta_2^{(i)}} \\
&= d_{\mathrm{ham}}\left(\theta_1,\theta_2\right) h^{\alpha-1+d} \ .
\end{aligned}
\tag{108}
$$

$\square$

**Private distributional tests and private Assouad method.**

**Lemma F.3** (Assouad's Lemma). *Assume that there exists $\tau > 0$ such that for any $\theta_1, \theta_2 \in \{0,1\}^N$,*

$$
d(S_{\theta_1}, S_{\theta_2})^2 \gtrsim d_{\mathrm{ham}}\left(\theta_1,\theta_2\right)\tau.
\tag{109}
$$

*Let $\hat{T}$ be any estimator and define $\hat{\theta} = \mathrm{argmin}_{\theta\in\{0,1\}^N} d(\hat{T}, \nabla\phi_\theta)$. Then it holds that*

$$
\sup_{\theta\in\{0,1\}^N} \mathbb{E}_{S_\theta}\left( \int \|\nabla\phi_\theta - \hat{T}\|^2 dP \right) \gtrsim \tau\sum_{i=1}^N \left( \mathbb{P}_{\theta_{-i}}(\hat{\theta}^i \neq 0) + \mathbb{P}_{\theta_{+i}}(\hat{\theta}^i \neq 1) \right),
\tag{110}
$$

*where $\mathbb{P}_{\theta_{+i}}$ and $\mathbb{P}_{\theta_{-i}}$ are the mixture distributions*

$$
\mathbb{P}_{\theta_{+i}} := \frac{1}{2^{N-1}}\sum_{\theta:\theta^{(i)}=1} S_\theta, \quad and \quad \mathbb{P}_{\theta_{-i}} := \frac{1}{2^{N-1}}\sum_{\theta:\theta^{(i)}=0} S_\theta \ .
\tag{111}
$$

*Note that in (110) there is a second layer or randomness that is implicit, and that is w.r.t. the estimator itself (for privacy for instance).*

*Proof.* The proof can be directly adapted from (Acharya et al., 2021) by substituting the notation with that used in the present paper. $\square$

**Lemma F.4.** *If $\hat{T}$ satisfies $\epsilon$-DP, then for any $i$,*

$$\mathbb{P}_{\theta_{-i}}(\hat{\theta}^i \neq 0) + \mathbb{P}_{\theta_{+i}}(\hat{\theta}^i \neq 1) \geq$$
$$\frac{1}{2^{N-1}} \sum_{\theta^1,\ldots,\theta^{i-1},\theta^{i+1}\ldots,\theta^N \in \{0,1\}} e^{-n\epsilon \mathrm{TV}\left(Q_{(\theta^1,\ldots,\theta^{i-1},0,\theta^{i+1}\ldots,\theta^N)},Q_{(\theta^1,\ldots,\theta^{i-1},1,\theta^{i+1}\ldots,\theta^N)}\right)} , \tag{112}$$

*where $\hat{\theta}$ is built from $\hat{T}$ as in Lemma F.3.*

*Proof.* This proof builds on coupling arguments from (Acharya et al., 2018; 2021; Lalanne et al., 2023a). Let us consider the coupling $\mathcal{C}$ that selects $\theta^1,\ldots,\theta^{i-1},\theta^{i+1}\ldots,\theta^N \in \{0,1\}$ uniformly at random, and then samples accordingly

$$X_1,\ldots,X_n \sim P \tag{113}$$

and

$$(Y_1,Y_1'),\ldots,(Y_n,Y_n') \sim Q \tag{114}$$

where $Q$ is the optimal coupling between $Q_{(\theta^1,\ldots,\theta^{i-1},0,\theta^{i+1}\ldots,\theta^N)}$ and $Q_{(\theta^1,\ldots,\theta^{i-1},1,\theta^{i+1}\ldots,\theta^N)}$. Here, the optimality means that if $(Y,Y') \sim Q$, then

$$\mathbb{P}(Y = Y') = 1 - \mathrm{TV}\left(Q_{(\theta^1,\ldots,\theta^{i-1},0,\theta^{i+1}\ldots,\theta^N)}, Q_{(\theta^1,\ldots,\theta^{i-1},1,\theta^{i+1}\ldots,\theta^N)}\right) . \tag{115}$$

The existence of such a coupling is well known (see, *e.g.* (Kallenberg, 1993)). Furthermore, we let $X_1',\ldots,X_n'$ be copies of $X_1,\ldots,X_n$.

Then $((X_1,\ldots,X_n,Y_1,\ldots,Y_n),(X_1',\ldots,X_n',Y_1',\ldots,Y_n'))$ (whose distribution is $\mathcal{C}$) is a coupling between $\mathbb{P}_{\theta_{+i}}$ and $\mathbb{P}_{\theta_{-i}}$.

Thus, we may write

$$\mathbb{P}_{\theta_{-i}}(\hat{\theta}^i \neq 0) + \mathbb{P}_{\theta_{+i}}(\hat{\theta}^i \neq 1) =$$
$$\mathbb{E}_{\mathcal{C}}\left(\mathbb{P}(\hat{\theta}^i \neq 0|X_1,\ldots,X_n,Y_1,\ldots,Y_n) + \mathbb{P}(\hat{\theta}^i \neq 1|X_1',\ldots,X_n',Y_1',\ldots,Y_n')\right) \tag{116}$$

where the inner randomness (i.e. inside the expectation) is only w.r.t. $\hat{\theta}^i$. Since $\hat{T}$ is $\epsilon$-DP, then so is $\hat{\theta}^i$ by post-processing.

Thus, by group privacy and the characteristic property of differential privacy,

$$\mathbb{P}_{\theta_{-i}}(\hat{\theta}^i \neq 0) + \mathbb{P}_{\theta_{+i}}(\hat{\theta}^i \neq 1) =$$

$$\mathbb{E}_{\mathcal{C}}\left(\mathbb{P}(\hat{\theta}^i((X_1,\dots,X_n,Y_1,\dots,Y_n)) \neq 0) + \mathbb{P}(\hat{\theta}^i((X_1',\dots,X_n',Y_1',\dots,Y_n')) \neq 1)\right)$$

$$\geq \mathbb{E}_{\mathcal{C}}\left(e^{-\epsilon d_{\mathrm{ham}}\left((X_1,\dots,X_n,Y_1,\dots,Y_n),(X_1',\dots,X_n',Y_1',\dots,Y_n')\right)}\mathbb{P}(\hat{\theta}^i((X_1',\dots,X_n',Y_1',\dots,Y_n')) \neq 0)\right.$$

$$\left. + \mathbb{P}(\hat{\theta}^i((X_1',\dots,X_n',Y_1',\dots,Y_n')) \neq 1)\right)$$

$$\geq \mathbb{E}_{\mathcal{C}}\left(e^{-\epsilon d_{\mathrm{ham}}\left((X_1,\dots,X_n,Y_1,\dots,Y_n),(X_1',\dots,X_n',Y_1',\dots,Y_n')\right)}\right.$$

$$\left.\underbrace{\left(\mathbb{P}(\hat{\theta}^i((X_1',\dots,X_n',Y_1',\dots,Y_n')) \neq 0) + \mathbb{P}(\hat{\theta}^i((X_1',\dots,X_n',Y_1',\dots,Y_n')) \neq 1)\right)}_{=1}\right)$$

$$= \mathbb{E}_{\mathcal{C}}\left(e^{-\epsilon d_{\mathrm{ham}}\left((X_1,\dots,X_n,Y_1,\dots,Y_n),(X_1',\dots,X_n',Y_1',\dots,Y_n')\right)}\right) \quad (117)$$

$$= \mathbb{E}_{\theta^1,\dots,\theta^{i-1},\theta^{i+1}\dots,\theta^N}\left(\right.$$

$$\mathbb{E}_{(X_1,\dots,X_n,Y_1,\dots,Y_n)(X_1',\dots,X_n',Y_1',\dots,Y_n')}\left(e^{-\epsilon d_{\mathrm{ham}}\left((X_1,\dots,X_n,Y_1,\dots,Y_n),(X_1',\dots,X_n',Y_1',\dots,Y_n')\right)}\right)\Big)$$

$$\overset{\mathrm{Jensen}}{\geq} \mathbb{E}_{\theta^1,\dots,\theta^{i-1},\theta^{i+1}\dots,\theta^N}\left(\right.$$

$$e^{-\epsilon\mathbb{E}_{(X_1,\dots,X_n,Y_1,\dots,Y_n)(X_1',\dots,X_n',Y_1',\dots,Y_n')}\left(d_{\mathrm{ham}}\left((X_1,\dots,X_n,Y_1,\dots,Y_n),(X_1',\dots,X_n',Y_1',\dots,Y_n')\right)\right)}\Big)$$

$$= \frac{1}{2^{N-1}}\sum_{\theta^1,\dots,\theta^{i-1},\theta^{i+1}\dots,\theta^N\in\{0,1\}} e^{-n\epsilon\mathrm{TV}\left(Q_{(\theta^1,\dots,\theta^{i-1},0,\theta^{i+1}\dots,\theta^N)},Q_{(\theta^1,\dots,\theta^{i-1},1,\theta^{i+1}\dots,\theta^N)}\right)},$$

where the last line comes from a simple computation. □

**Conclusion of the proof.** As a consequence of Lemma F.1, Lemma F.2, Lemma F.3 and Lemma F.4,

$$\inf_{\hat{T}}\sup_{\theta\in\{0,1\}^N}\mathbb{E}_{S_\theta}\left(\int\|\nabla\phi_\theta - \hat{T}\|^2 dP\right) \gtrsim Nh^{2\alpha+d}e^{-Cn\epsilon h^{\alpha-1+d}}, \quad (118)$$

where $C > 0$ is a positive constant. Finally, since $N \asymp h^{-d}$, taking $h \asymp (n\epsilon)^{-\frac{1}{\alpha-1+d}}$ yields

$$\inf_{\hat{T}}\sup_{\theta\in\{0,1\}^N}\mathbb{E}_{S_\theta}\left(\int\|\nabla\phi_\theta - \hat{T}\|^2 dP\right) \gtrsim (n\epsilon)^{-\frac{2\alpha}{\alpha-1+d}}, \quad (119)$$

and (94) then concludes the proof.

# G. Practical algorithm

This section provides further details on how to obtain a numerically tractable algorithm derived from the ideas presented in Section 3 and Section 4.

### G.1. Choosing $J$

First, we determine $J$ to achieve the best bias-variance tradeoff. Using Equation (24), we set

$$J^* = \min\left(\left\lceil\frac{\log_2(n)}{d-2+2\alpha}\right\rceil, \left\lceil\frac{\log_2(n\epsilon)}{d+2\alpha}\right\rceil\right).$$

This choice asymptotically yields an error

$$\lesssim J^* R^2 (\log_2(n) + \log_2(n\epsilon)) \times \text{Upper-Bound Equation (25)},$$

where we explicitly track the scaling in $R$ and retain logarithmic terms.

### G.2. Constructing the Covering & Enforcing Conditions

With $J^*$ fixed, we construct the covering. From Equation (39), controlling the infinite functional norm by the vector's infinite norm requires computing $\sqrt{\text{Vol}(\tilde{\Omega})}$. In the paper, we assume $\text{Vol}(\tilde{\Omega}) = 3^d$, but since $\tilde{\Omega}$ only needs to be a hypercube containing $\Omega$ in its interior, we can instead set

$$\text{Vol}(\tilde{\Omega}) = (1 + \gamma)^d$$

for any fixed $\gamma > 0$.

In Lemma 4.2 and Theorem 4.3, we first construct a $\delta = C/(n\epsilon)$ covering in vector infinite norm for the space of wavelet coefficients up to $J^*$, within a radius $CM^2$, where

$$C = 2\sqrt{\text{Vol}(\tilde{\Omega})}.$$

This covering is straightforward since it requires only axis-wise discretization in the space of wavelets coefficients.

The challenging part follows: the conditions on Hessian eigenvalues and the potential's norm/gradient may not hold. We enforce them via the reasoning in the end of Appendix E.2. While this step is theoretically sound for obtaining the upper bound, we believe it is intractable in practice.

To address this, we can approximate using a grid, as in Section 6. This does not affect the privacy analysis but may introduce numerical errors that vanish as the discretization step decreases. Candidate potentials are now represented by their grid discretization and live in the span of wavelet discretizations up to level $J^*$.

For each candidate in the initial covering, we check whether a nearby potential (within distance $\delta$) satisfies the conditions on the Hessian, gradient, and potential itself, following Definition 2.3 and the end of Appendix E.2 in a discretized manner. At this stage, existence testing reduces to a convex optimization problem—minimizing the infinite norm of a vector under linear constraints, since numerical differentiation schemes are linear operators. A numerical solver can efficiently handle this step. The set $C_{J,M}$ is thus constructed using this quasi-projection approach (see Appendix E.2) applied to each candidate potential from the initial covering.

### G.3. Conclusion

This concludes the practical implementation of our estimator. While computationally expensive, it was designed for theoretical purposes. However, we emphasize that the selection rule in Section 3 remains valid for any set of convex potentials with bounded infinite norm—ensuring meaningfulness in optimal transport theory and privacy analysis.

The complexity mainly lies in generating candidate potentials (which is not due to privacy but rather to the difficulty of the non-parametric nature of the potential estimation as in (Hütter & Rigollet, 2021)), whereas the selection procedure itself is more practical. Thus, with a realistic potential generator (e.g., leveraging prior knowledge), our private selector should yield strong utility in practice.

## H. Potential alternative approaches

### H.1. Plugin estimators

An alternative approach could be to estimate privately both the input and output distributions, and to consider the optimal transport between those estimates. Such a technique is usually referred to as a *plugin* estimator.

For our problem, it typically requires additional assumptions on the regularity of the source or the target distributions (Manole et al., 2024). We are confident that such a technique could lead to good results under this more restrictive setup, yet it would require extra work as it requires stability of the estimates in $W_2$ distance, whereas most results on density estimation under differential privacy are stated either in terms of infinite or $L_2$ norms (Wasserman & Zhou, 2010; Barber & Duchi, 2014; Lalanne et al., 2023b; Lalanne & Gadat, 2024).

## H.2. Gradient-based approaches

The semi-dual problem is a finite-dimensional convex optimization problem once discretized in the space of wavelet coefficients. Differentially private algorithms for convex optimization (such as DP-SGD) could, in principle, be applied directly.

However, the problem is that $f^*$ (the Fenchel conjugate which appears in the optimization problem) does not admit a closed-form expression in terms of the coefficients, akin to its gradient. In addition, it is not guaranteed that the problem benefits from gradient smoothness or strong convexity, also limiting approaches such as objective perturbation.

