# OpenReview forum: "On the Private Estimation of Smooth Transport Maps"
_ICML.cc/2025/Conference — ICML 2025 poster_

### Official Review · Reviewer_FYXP · 2025-03-06

**Overall Recommendation:** 3

**Summary:**

This paper leverages Differential Privacy (DP) to address concerns about privacy leakage when estimating the transport map between two distributions derived from user data. The authors propose a DP transport map estimator and establish its statistical guarantees in terms of an upper bound and a minimax lower bound.

**Claims And Evidence:**

The authors establish statistical guarantees for the proposed private transport map estimator. However, there is a lack of insight and clarification regarding these bounds, particularly in terms of how they compare to the non-DP version, and how changes in $\epsilon$ affect the convergence rate and the estimation's optimality. The theoretical claim could be strengthened by deeper insights and more detailed explanations.

**Essential References Not Discussed:**

NA

**Experimental Designs Or Analyses:**

The performance of the privately estimated map compared to the true map is shown in Figure 2. Although the estimated map visually appears close to the ground truth, there is no quantitative measure to assess the estimation error and confirm its optimality.

The figures appear to show results from only a single run of the algorithm, but a single run provides little insight into the algorithm's general performance and variance. What is its average approximation error across multiple simulations? The fact that this is neither displayed nor mentioned is quite concerning.

To address privacy concerns, it could be interesting to evaluate the performance of the estimator in cases where the underlying distribution is heavily skewed.

**Methods And Evaluation Criteria:**

The problem addressed in this paper is important. However, the method is not novel; it is a routine development that combines Differential Privacy (DP) with existing transport map estimators.

**Other Comments Or Suggestions:**

NA

**Other Strengths And Weaknesses:**

NA

**Questions For Authors:**

My main concern is with the numerical experiments. Please see the comments under 'Experimental Design and Analysis'.

**Relation To Broader Scientific Literature:**

The main results of this paper could contribute to applications and methodologies involving optimal transport maps using real user data.

**Theoretical Claims:**

There are no red flags in the theoretical part, but there is a possibility that I may have missed something.

---

> ### Author Rebuttal · Authors · 2025-03-31
>
> Dear Reviewer,
>
> We sincerely appreciate the time and effort you have taken to review our paper. Below, we provide detailed responses to each of your comments.
>
> - **Comment 1:** *The authors establish statistical guarantees for the proposed private transport map estimator. However, there is a lack of insight and clarification regarding these bounds, [...] and more detailed explanations.*
>
>   **Response:** We would like to point out that we included elements addressing this question between lines 359 and 362, where we exhibit the regime in which any private algorithm, regardless of the specifics, will necessarily experience performance degradation. We will enhance the discussion after Theorems 4.3 and 5.1 to better explain the bounds in different regimes. For instance, our upper bound recovers the usual optimal rate of estimation in the limit $\varepsilon \to \infty$ (i.e., no privacy). In the regime of high smoothness ($\alpha \to \infty$), we can see that a constant privacy budget does not affect utility. In intermediate regimes, we will ensure to detail this point using the inversion formula.
>
> - **Comment 2:** *However, the method is not novel; it is a routine development that combines Differential Privacy (DP) with existing transport map estimators.*
>
>   **Response:** Our work indeed combines existing transport map estimators with an established Differential Privacy mechanism, but its statistical analysis requires a careful link between these two components, which we believe results in a valuable contribution. The slight mismatch between our upper and lower bounds (which does not exist in non-private transport map estimation) arises as a direct consequence of the additional challenges posed by privacy constraints. A significant portion of the differential privacy literature consists of applying standard mechanisms (e.g., Gaussian, Exponential, etc.) to prescribed quantities, but the challenge lies in quantifying the utility of such procedures.
>
> - **Comment 3:** *The performance of the privately estimated map compared to the true map is shown in Figure 2. Although the estimated map visually appears close to the ground truth, there is no quantitative measure to assess the estimation error and confirm its optimality & The figures appear to show results from only a single run [...]*
>
>   **Response:** Our work is primarily theoretical in nature, and the experiments are mainly illustrative of the soundness of our approach. We will add a numerical figure of merit that measures the estimation error (e.g., the $L^2$ error between the two maps as a function of $n$). We can also include the standard deviation when averaging over multiple runs. However, we want to emphasize that the experiments serve to demonstrate that the proposed method is effective, at least in a toy example. The main contribution of the paper is to establish the first statistical bound for the problem at hand.
>
> - **Comment 4:** *To address privacy concerns, it could be interesting to evaluate the performance of the estimator in cases where the underlying distribution is heavily skewed.*
>
>   **Response:** We are unsure of what the reviewer means by "skewed" in this context. If "skewed" refers to cases where some regions have a high probability mass while others are nearly empty, this is indeed an interesting question. However, this scenario is excluded by our assumptions (see Definition 2.1), as it induces instabilities in the estimation procedure, even in the absence of privacy constraints. We believe this is a valuable question for future empirical research, but it is outside the scope of the current paper.
>
> We would be happy to discuss any remaining questions during the author-reviewer discussion phase.

---

### Official Review · Reviewer_KvNT · 2025-03-11

**Overall Recommendation:** 3

**Summary:**

This paper presents a differentiably private (DP) estimator for continuous $W_2$ transport maps, obtained from discrete samples from source and target distributions. It builds on top of the framework established by [Hutter & Rigollet 21] for estimating the Brenier potential ($\nabla f$ returns the transport map), via wavelet approximation. Privacy is achieved with a noisy (Laplace) report argmin mechanism, applied to a covering of the admissible family of potentials. They are the first to provide upper and lower bounds for such a DP problem (in terms of $L^2$ error on the transport plan), but optimality of such bounds is not proven. The authors also present a simple practical implementation for a planar scenario that leverages a grid-based discretization.

## Update after rebuttal

I appreciate the clarifications from the authors, but remain at a weak accept, especially as some of my main concerns were shared by other reviewers.

**Claims And Evidence:**

The paper is primarily theoretical, so the need for empirical evidence is minimal. Detailed proofs are provided for the claims given, that mostly leverage prior analytical tools from [Hutter & Rigollet 21] and established mechanisms from the DP literature.

**Essential References Not Discussed:**

None: see above.

**Experimental Designs Or Analyses:**

The experiment proposed is minimal, but sufficient.

**Methods And Evaluation Criteria:**

Yes, methods are fine. The experiment is small, but that is fine, as the focus is on the analysis and statistical bounds.

I could add that it would be nice to see which components of the bounds seem to dominate in which regimes over particular dimensions, sample sizes, etc. Not vital, in my opinion, however.

**Other Comments Or Suggestions:**

Explanation of rating: I found the paper to provide an interesting look at a heretofore unexpored problem. I did not give a higher rating than borderline accept, as it was unclear to me if the method is practically useful. For the one example displayed in Figure 2, I was surprised to see the degree of disagreement, given the use of $n=200000$ points. Additionally, I was a bit unsure of the degree of novelty in the theoretical construction, given that it builds so strongly off of a typical DP mechanism and the work of [Hutter and Rigollet 21].

**Other Strengths And Weaknesses:**

A main strength of the paper is its clarity. I found the exposition to do a great job presenting and outlining their basic technique and results.

If I had to level one clarity complaint, it's that I would have included a few words about the construction of the finite covering family, namely how one maintains convexity of the functions.

Also, for reproduction's sake, it would be quite nice to include an implementation of the practical algorithm for the research community.

**Questions For Authors:**

See the one clarity weakness that I had a problem with above.

**Relation To Broader Scientific Literature:**

The paper is the first to provide some statistical guarantees on DP OT estimation in the continuous setting. The related work does a good job of establishing this and referencing relevant work (or at least, I did not notice any glaring omissions).

**Theoretical Claims:**

I did not have time to give a detailed check for the theoretical proofs in the appendices, but the arguments in the main text were correct.

---

> ### Author Rebuttal · Authors · 2025-03-31
>
> Dear Reviewer,
>
> We sincerely appreciate the time and effort you have taken to review our paper. Below, we provide detailed responses to each of your comments.
>
> - **Comment 1:** *I could add that it would be nice to see which components of the bounds seem to dominate in which regimes over particular dimensions, sample sizes, etc. Not vital, in my opinion, however.*
>
>   **Answer:** We thank you for the suggestion. Since this point was raised in other reviews, we propose to extend the discussion in the relevant paragraph. For instance, our upper bound recovers the usual optimal rate of estimation in the limit $\varepsilon \to \infty$ (i.e., no privacy). In the regime of high smoothness ($\alpha \to \infty$), we can see that a constant privacy budget does not affect utility. In intermediate regimes, we will ensure to detail this point using the inversion formula.
>
> - **Comment 2:** *If I had to level one clarity complaint, it's that I would have included a few words about the construction of the finite covering family, namely how one maintains convexity of the functions.*
>
>   **Answer:** The construction of the covering (detailed in Section E.2) maintains convexity via the algorithm developed in lines 1194-1200. Alternatively, since the set of potentials of interest is closed and convex in a Euclidean space, it is possible to replace the presented procedure with a projection step, which is contractive and would allow us to gain a factor of $2$ in the constant.
>   We emphasize that this construction mainly makes sense from a theoretical point of view in order to obtain the statistical upper bound. In practice, this construction can be challenging, and we recommend checking the convexity of the candidate potentials beforehand.
>
> - **Comment 3:** *Also, for reproduction's sake, it would be quite nice to include an implementation of the practical algorithm for the research community.*
>
>   **Answer:** We thank the reviewer for the suggestion. We will include the link to the code on our GitHub in the final version of this article.
>
> - **Comment 4:** *Explanation of rating: I found the paper to provide an interesting look at a heretofore unexplored problem. I did not give a higher rating than borderline accept, as it was unclear to me if the method is practically useful. For the one example displayed in Figure 2, I was surprised to see the degree of disagreement, given the use of $n=200000$ points. Additionally, I was a bit unsure of the degree of novelty in the theoretical construction, given that it builds so strongly off of a typical DP mechanism and the work of [Hutter and Rigollet 21].*
>
>   **Answer:** The degree of disagreement in Figure 2 is an artifact of the evaluation rather than of the estimation. We will update the figure with more precise evaluation and add figures of merit for the disagreement. Since the empirical evaluation was also pointed out by other reviewers, we plan to add to the final version a figure representing the error as a function of $n$, the sample size, as well as error bars.
>
>   Regarding the degree of novelty in the theoretical construction, we refer to our response to Reviewer tgzq. For instance, the additional error related to privacy has to be precisely controlled in the estimation; this is the purpose of Lemma 2.5, which is crucial to linking smooth map estimation and the differentially private component.
>
>   When it comes to practical applicability, we believe that our method should be split into two parts: an exploration phase and a selection phase. The exploration phase (the covering construction) can be challenging for practical problems. However, the selection phase (via the empirical semi-dual criterion) is perfectly applicable in practical scenarios where a prescribed set of candidate potentials has been suggested beforehand (e.g., via prior information).
>
> We would be happy to discuss any remaining questions during the author-reviewer discussion phase.

---

### Official Review · Reviewer_3Ra6 · 2025-03-13

**Overall Recommendation:** 3

**Summary:**

This paper considers the problem of learning the optimal transport map from $P$ to $Q$, given samples from each, under the requirement of (pure) differential privacy. They impose assumptions on smoothness of the optimal map which are standard in the map estimation literature. They obtain an upper bound using a noisy argmin mechanism and a covering argument. Their lower bound employs standard packing arguments.

## Update after Rebuttal

I am still borderline. The result is relatively interesting and the domain is new from the perspective of DP, but the privacy tools applied are pretty standard. Use of standard privacy tools is fine - the real question is whether their analysis is novel/significant enough for ICML. I am not sure - the covering and sensitivity analysis seem somewhat routine. If the accompanying lower bound was tight, then it wouldn't be fair to complain about this, but there is still a gap. On the other hand, I do not doubt correctness, and the paper was well-written and nice to read.

**Claims And Evidence:**

Yes, this is primarily a theoretical paper, with results and proofs that appear sound.

**Essential References Not Discussed:**

These works look at the problem of private distribution estimation under Wp and were not included. I think they have tight rates and some efficient algorithms.

Algorithmically Effective Differentially Private Synthetic Data
Yiyun He, Roman Vershynin, Yizhe Zhu
COLT 2023

Private Measures, Random Walks, and Synthetic Data
March Boedihardjo, Thomas Strohmer, and Roman Vershynin
Probability theory and related fields, 2024

**Experimental Designs Or Analyses:**

The experiments are naturally limited to 2d because of poor computational scaling with dimension. However, I think a statistical contribution is sufficient given that this problem has not been explored before. Plus, when smoothness is low, even the statistical rates without privacy scale poorly. (On the other hand, maybe that could enable "efficient" algorithms that scale polynomially in the sample size, since that is already exponential in dimension)

**Methods And Evaluation Criteria:**

They employ the standard evaluation metric for the map estimation task. The technical tools for designing the upper and lower bounds are standard in the literature (but have not yet been applied to this domain)

**Other Comments Or Suggestions:**

I'm sure there is a simple reason why you use the extended domain $\tilde{\Omega}$ over $\Omega$, but I didn't see any discussion. Could you be explicit about that?

It would be nice if the proof of Lemma 2.5 was split into a lemma capturing the needed argument from Hütter & Rigollet and then your own contribution.

**Other Strengths And Weaknesses:**

I think the novelty/significance of these contributions is a bit borderline. Since this problem has not been studied before, I am definitely open to acceptance. However, the techniques for the upper and lower bounds are all very standard in the literature, and I don't think this problem has not been studied due to its difficulty, but rather because it is a bit niche. To calibrate, I would probably accept this paper at a conference like AISTATS but reject for a conference like COLT.

I think matching upper and lower bounds, or clear answers to my questions below, would convince me to accept.

**Questions For Authors:**

Can you resolve the complexity of this problem in 1D? Here the connection to quantile functions should considerably simplify the problem.

What about first doing private density estimation to obtain estimates $\hat{P}$ and $\hat{Q}$ which are accurate under $W_2$, then computing a smooth map for the $W_2(\hat{P},\hat{Q})$ problem (which is private by post-processing)? Analysis isn't obvious to me, but is there anything you can say here? To be fair, I don't think private density estimation has been well-studied under smoothness.

**Relation To Broader Scientific Literature:**

They provide good background on the map estimation literature. There is a related problem of privately estimating a distribution/density under Wp. They mention some work in this direction but I note a couple missing references below.

**Theoretical Claims:**

I skimmed through all of the appendix proofs, but did not check them line by line. However, these methods are super natural to apply here so I would be very surprised if there are any major issues.

---

> ### Author Rebuttal · Authors · 2025-03-31
>
> Dear Reviewer,
>
> We sincerely appreciate the time and effort you have taken to review our paper. Below, we provide detailed responses to each of your comments.
>
> - **Comment 1:** *These works look at the problem of private distribution estimation under Wp and were not included. I think they have tight rates and some efficient algorithms.*
>
>   **Answer:** We thank you for the suggested references and will include them in the relevant body of literature. After reviewing them, they unfortunately do not seem to exploit smoothness, which means they will probably not lead to better rates for our problem. However, it is possible that their numerical performance is stronger than that of the presented method.
>
> - **Comment 2:** *I think the novelty/significance [...]  conference like COLT.*
>
>   **Answer:** Yes, the problem may seem a bit niche, but it still has some practical interest. The estimation of transport maps plays an important role in practical applications. Ensuring the privacy of such transport maps is particularly beneficial in applications involving the infringement of fundamental rights, such as bias analysis in machine learning algorithms. For instance, in [1] and [2], OT maps enable counterfactual analysis of discrimination. For a discussion about matching upper bounds, see our comment about the use of density estimators later in our response.
>
> - **Comment 3:** *I'm sure there is a simple reason why you use the extended domain $\tilde{\Omega}$ over $\Omega$, but I didn't see any discussion. Could you be explicit about that?*
>
>   **Answer:** In a nutshell, this is a consequence of estimating the transport map through smooth Brenier potentials: as the transport map is constructed as the gradient of the potential $T=\nabla f$, $f$ must be defined on a larger domain than $\Omega$ so that the gradient exists everywhere there. Thus, this is not linked to differential privacy but rather to transport map estimation through potentials. In fact, $\tilde{\Omega}$ could be chosen arbitrarily as long as $\Omega$ is included in its interior and it is a hypercube. The hypercube constraint arises from the support adequation condition described in Appendix A.
>
> - **Comment 4:** *It would be nice if the proof of Lemma 2.5 was split into a lemma capturing the needed argument from Hütter & Rigollet and then your own contribution.*
>
>   **Answer:** In the present proof, we try to clarify what is relevant for privacy and what simply follows the steps of Hütter & Rigollet. Unfortunately, since their result uses more advanced concentration tools than a uniform bound, we did not find a way to split the lemma in a way similar to a triangular inequality. Instead, we had to track the propagation of suboptimality in their proof and make the necessary measurability assumptions. We can add a remark after the result detailing which terms come from Hütter and Rigollet and which terms pertain to privacy.
>
> - **Comment 5:** *What about first doing private density estimation to obtain estimates $\hat{P}$ and $\hat{Q}$ that are accurate under $W_2$, then computing a smooth map for the $W_2(\hat{P}, \hat{Q})$... well studied under smoothness.*
>
>   **Answer:** Indeed, the approach proposed by the reviewer is sound. In fact, private density estimation has even been studied under smoothness (see [5]) for the $L^2$ loss with optimal rates. Furthermore, [4] demonstrated that density estimators can be used as suggested if they are sufficiently precise in $W_2$ distance. However, two problems arise: (i) the estimation must be in $W_2$ distance and not $L^2$, though we are confident that this should not be a problem as the techniques appear to be manageable with privacy, and (ii) it requires extra smoothness assumptions on the source and target distributions (on top of those on the map), limiting the applicability of the method. While this method is likely to yield optimal rates (at least in the one-sample setting where $P$ is known), we aimed to study the problem under the purest set of hypotheses. We can add a remark about this direction in the article.
>
> - **Comment 6:** *Can you resolve the complexity of this problem in 1D? Here, the connection to quantile functions should considerably simplify the problem.*
>
>   **Answer:** In 1D, it should be possible to import results from quantiles/CDFs since the transport map is known, but it is unclear whether this will be optimal.
>
> We would be happy to discuss any remaining questions during the author-reviewer discussion phase.
>
> [1] Transport-based counterfactual models, De Lara et al., JMLR 2024
> [2] Fliptest: Fairness testing via optimal transport, Black et al., CFAT 2020
> [3] Plugin estimation of smooth optimal transport maps, Manole et al., Annals of Stats., 2024
> [4] Discrete Wasserstein barycenters: Optimal transport for discrete data, Anderes et al., Mathematical Methods of Operations Research 2016
> [5] Privately Estimating Smooth Distributions on the Hypercube by Projections, Lalanne et al., ICML 2024

---

### Official Review · Reviewer_tgzq · 2025-03-18

**Overall Recommendation:** 3

**Summary:**

The paper studies the private estimation of smooth optimal transport maps using a semi-dual formulation of optimal transport, where the transport map is obtained as the gradient of a Brenier potential. To make the problem tractable, they restrict the function space using standard nonparametric estimation techniques with wavelet bases, following the approach of Hütter & Rigollet, which provides minimax bounds for transport map estimation. The core contribution is the introduction of differential privacy (DP) by injecting Laplace noise into the empirical semi-dual objective function $\hat S(f_i)$ ensuring that the estimation process remains private while preserving statistical guarantees. To implement this, they define a finite set of candidate functions $C_{J, M}$ which forms a discrete covering of the function space $V_J$ (merely a span of restricted wavelet basis) and the semi-dual objective is then computed for each function $f_i$​ in this set, and added classical Laplace mechanism to the values to enforce privacy.
The private transport map is then obtained as the gradient of the selected function. The paper rigorously analyzes the trade-off between privacy, statistical accuracy, and discuss some computational feasibility, proving upper and lower bounds on the estimation error and demonstrating that privacy constraints degrade the convergence rate but remain reasonable. Unfortunately, the lower and upper bound does not match.
Some numerical experiments provide some illustration on toy examples.

**Claims And Evidence:**

The claims of the paper are mostlty theoretical and supported by formal proof.

**Essential References Not Discussed:**

No

**Experimental Designs Or Analyses:**

As commented above, the numerical experiments section is rather weak and serve as toy illustrations for the theoretical claims.

**Methods And Evaluation Criteria:**

Only toy experiments are presented to illustrate how the proposed transport map are close to the optimal one. Unfortunately, these synthetic illustrations might not reflect practical performances, specially for nowadays highdimensional ML problems.

**Other Comments Or Suggestions:**

Several use of same notation $M$ but with different meaning accross the text

**Other Strengths And Weaknesses:**

The theoretical analysis and presentations are cleanly presented with transparent assumptions.
The minimax bounds is similar to (Hutter & Rigollet, 2021) $n^{-1} \vee n^{\frac{-2 \alpha}{2 \alpha - 2 + d}}$ with the additional differentially private term $(n \epsilon)^{ \frac{-2\alpha}{2\alpha + d} }$.

This additional term is unfortunately not tight; how the privacy mechanism interacts with function approximation is quite mysterious at some points.

**Questions For Authors:**

- Assuming the function class $\mathcal{F}$ is parameterized in a finite-dimensional space and Once the problem is formulated using the empirical distributions, the semi-dual problem becomes a finite-dimensional convex optimization problem. Differentially Private (DP) algorithms for convex optimization already exist and could, in principle, be applied directly? Examples in mind (https://arxiv.org/abs/2207.08347 , https://jmlr.org/papers/volume12/chaudhuri11a/chaudhuri11a.pdf , https://proceedings.mlr.press/v23/kifer12.html).

- Any plan to include experiments with real dataset or more realistic ML setting where DP OT map are needed?

- The choice of the constant $J$ and the finite function set $C_{J, M}$ ​ remains unclear throughout the paper. It is difficult to keep track of how these quantities are selected, particularly since $J$ appears to depend on the dimension $d$ in an implicit manner, yet its exact scaling is not clearly derived or discussed. Additionally, the "constant" $R$ is not formally defined (coudln't find it), making it challenging to understand its role in the function space discretization. Theorem 4.3, which should provide precise details on these constants, remains evasive on their explicit dependence, even in the proof. This lack of clarity significantly affects the practical implementability of the proposed method


 "## update after rebuttal"

I thanks the authors for the clarifications they provided during the discussions. The points on selecting the parameters are a bit more transparent. I maintain my score and not increase it because the numerical experiments are clearly weak and do not illustrate the initial motivation presented by the authors.

**Relation To Broader Scientific Literature:**

N.A

**Theoretical Claims:**

I briefly skim into the appendix but not read in details.

---

> ### Author Rebuttal · Authors · 2025-03-31
>
> Dear Reviewer,
>
> We sincerely appreciate the time and effort you have taken to review our paper. Below, we provide detailed responses to each of your comments.
>
> - **Comment 1:** *Only toy experiments are presented to illustrate how the proposed transport maps are close to the optimal ones. Unfortunately, these synthetic illustrations might not reflect practical performance, especially for modern high-dimensional ML problems. & Any plans to include experiments with real datasets or more realistic ML settings where DP OT maps are needed?*
>
>   **Response:** The experiments in the paper are primarily intended to illustrate the theoretical soundness of our approach. Specifically, the private estimator we consider has the advantage of yielding tractable statistical bounds. However, it may not be the most efficient choice from a numerical standpoint. This observation extends to minimax estimation of transport maps in general (even without privacy constraints), where entropic approximation performs best in practice but remains theoretically challenging. Exploring this further is an interesting direction for future research. To clarify the practical scope of our work, private maps can be used in fairness applications, for measuring population flux, and in other domains where transport maps may contain sensitive information on individuals.
>
> - **Comment 2:** *The theoretical analysis and presentation are clean, with transparent assumptions. The minimax bounds resemble those in (Hutter & Rigollet, 2021) with an additional differentially private term. However, this additional term is not tight, and the interaction between the privacy mechanism and function approximation remains somewhat unclear.*
>
>   **Response:** Privacy introduces an extra variance term, the magnitude of which depends on the dimension of the function approximation space. This same space also controls the bias of the estimation procedure, with its dependence tuned by smoothness. Consequently, the dimension of the functional space serves as a cutoff parameter that balances the bias-variance tradeoff. We will expand the discussion following Theorem 4.3 to clarify the different regimes of the bound and how the private bound differs from the classical case.
>
> - **Comment 3:** *Assuming the function class $\mathcal{F}$ is parameterized in a finite-dimensional space, and once the problem is formulated using empirical distributions, the semi-dual problem becomes a finite-dimensional convex optimization problem. Differentially private (DP) algorithms for convex optimization already exist and could, in principle, be applied directly.*
>
>   **Response:** Indeed, as noted, the problem is convex in $f$ and Lipschitz, and we have explored this direction. However, a key difficulty arises because $ f^* $ (the Fenchel conjugate appearing in the optimization problem) does not have a closed-form expression in terms of the coefficients, nor does its gradient. Additionally, we were unable to establish gradient smoothness or strong convexity, which excludes the suggested references. Furthermore, the absence of strong convexity means that DP-SGD techniques are not in their most favorable cases. Adding regularization could improve this but such biased techniques fall out of the scope of this article. That said, since the objective function is convex and Lipschitz-continuous (see Eq. (40), where we establish a Lipschitz constant scaling as $2^{Jd/2}$), it is almost everywhere differentiable. This suggests that gradient-based optimization might be possible, (although computing $\nabla_{\text{coeffs}} f^*$  remains challenging). Whether such an approach would yield a favorable utility-privacy tradeoff remains an open question.
>
> - **Comment 4:** *The choice of the constant $J$ and the finite function set $C_{J,M}$ remains unclear throughout the paper. It is difficult to track how these quantities are selected, particularly since $J$ appears to depend implicitly on the dimension $d$ without an explicit derivation. Additionally, the "constant" $R$ is not formally defined (I couldn't find it), making it challenging to understand its role in function space discretization. Theorem 4.3, which should clarify these constants, does not explicitly detail their dependence, even in the proof. This lack of clarity significantly impacts the practical implementability of the proposed method.*
>
>   **Response:** The introductory paragraph of Section 2 clarifies what is considered a constant and what is not. Specifically, we precisely track all factors involving $J, n, \varepsilon$. The parameter $R$ controls the level of $\alpha$-smoothness, as defined in Equation (4). We will revise the relevant paragraph to make this clearer. For practical applications, as discussed in Section 6, we recommend selecting a set of candidate potentials and using our method as a selection tool.
>
> We would be happy to discuss any remaining questions during the author-reviewer discussion phase.

---

> > ### Comment · Reviewer_tgzq · 2025-04-02
> >
> > Thanks for the comments and clarifications.
> >
> > > To clarify the practical scope of our work, private maps can be used in fairness applications, for measuring population flux, and in other domains where transport maps may contain sensitive information on individuals.
> >
> > Yes, it would be nice to showcase experiments for the problem you mentioned being able to solve. That being said, I agree that the paper is theory perfumed, I am not requesting those experiments for acceptance.
> >
> > > The introductory paragraph of Section 2 clarifies what is considered a constant and what is not.
> >
> > Still there is several things to track in the paper and as of now, I would struggle to implement the proposed method based on the presentation of the paper. So please provide more guidance on that.

---

> > > ### Author Response · Authors · 2025-04-04
> > >
> > > We thank the reviewer for the quick response, allowing us time to provide a detailed answer. Below, we clarify the practical construction and algorithm.
> > >
> > > ## Choosing $J$
> > > First, we determine $J$ to achieve the best bias-variance tradeoff. Using Equation (24), we set
> > > $$
> > > J^* = \min \Big( \Big\lceil \frac{\log_2(n)}{ d - 2 + 2 \alpha} \Big\rceil, \Big\lceil\frac{\log_2(n \epsilon)}{ d + 2 \alpha}\Big\rceil \Big).
> > > $$
> > > This choice asymptotically yields an error
> > > $$
> > > \lesssim J^* R^2 (\log_2(n ) + \log_2(n \epsilon)) \times \text{Upper-Bound Equation (25)},
> > > $$
> > > where we explicitly track the scaling in $R$ and retain logarithmic terms.
> > >
> > > ### Constructing the Covering
> > > With $J^*$ fixed, we construct the covering. We acknowledge that the current presentation requires backtracking through proofs and that some constants are implicit. We will explicitly detail the construction, either in the main text or in an appendix, depending on length constraints.
> > >
> > > From Equation (39), controlling the infinite functional norm by the vector's infinite norm requires computing $\sqrt{\text{Vol}(\tilde{\Omega})}$. The article assumes $\text{Vol}(\tilde{\Omega}) = 3^d$, but since $\tilde{\Omega}$ only needs to be a hypercube containing $\Omega$ in its interior, we can instead set
> > > $$
> > > \text{Vol}(\tilde{\Omega}) = (1+\gamma)^d
> > > $$
> > > for any fixed $\gamma > 0$.
> > >
> > > ### Enforcing Conditions
> > > In Lemma 4.2 and Theorem 4.3, we first construct a $\delta = C / (n \epsilon)$ covering in vectorial infinite norm for the space of wavelet coefficients up to $J^*$, within a radius $C M^2$, where
> > > $$
> > > C = 2 \sqrt{\text{Vol}(\tilde{\Omega})}.
> > > $$
> > > This covering is straightforward since it requires only axis-wise discretization in the space of wavelets coefficients.
> > >
> > > The challenging part follows: the conditions on Hessian eigenvalues and the potential’s norm/gradient may not hold. We enforce them via the reasoning in lines 1194-1198. While this step is theoretically sound for obtaining the upper bound, we believe it is intractable in practice.
> > >
> > > ### Grid Approximation
> > > To address this, we can approximate using a grid, as in Section 6. This does not affect privacy analysis but may introduce numerical errors that vanish as the discretization step decreases. Candidate potentials are now represented by their grid discretization and live in the span of wavelet discretizations up to level $J^*$.
> > >
> > > For each candidate in the initial covering, we check whether a nearby potential (within distance $\delta$) satisfies the conditions on the Hessian, gradient, and potential itself, following Definition 2.3 and lines 1194-1198 in a discretized manner.
> > >
> > > At this stage, existence testing reduces to a convex optimization problem (without privacy because the covering construction is agnostic to data)—minimizing the infinite norm of a vector under linear constraints, since numerical differentiation schemes are linear operators. A numerical solver can efficiently handle this step.
> > >
> > > The set $C_{J, M}$ is thus constructed using this quasi-projection approach (lines 1194-1198) applied to each candidate potential from the initial covering.
> > >
> > > ## Conclusion
> > > This concludes the "practical" implementation of our estimator. While computationally expensive, it was designed for theoretical purposes. However, we emphasize that the selection rule in Section 3 remains valid for any set of convex potentials with bounded infinite norm—ensuring meaningfulness in optimal transport theory and privacy analysis.
> > >
> > > The complexity mainly lies in generating candidate potentials (which is not due to privacy but rather to the difficulty of the non-parametric nature of the potential estimation as in [Hutter&Rigollet,2021]), whereas the private selection procedure itself is more practical. Thus, with a realistic potential generator (e.g., leveraging prior knowledge), our private selector should yield strong utility in practice.

---

### Decision · Program_Chairs · 2025-05-01

**Decision:**

Accept (poster)

**Comment:**

This paper tackles the theoretically important and previously underexplored problem of estimating smooth optimal transport maps under differential privacy constraints. The authors propose a private estimator using a covering argument and establish upper and lower bounds on its statistical performance. The work is clearly written, with clean formalism and a sound analytical framework. Reviewers appreciated the clarity of the presentation, the relevance of the problem, and the fact that this is the first paper to provide nontrivial statistical guarantees in this setting. That said, several reviewers raised concerns about the limited practical applicability of the proposed estimator. The algorithm, while theoretically motivated, is computationally expensive and not easily implementable based on the current description. Additionally, the theoretical contributions, while solid, rely on fairly standard tools from the literature, and do not fully resolve important questions such as tightness of the bounds or efficient estimation. The experimental section is minimal, providing only toy illustrations, and does not convincingly demonstrate the utility of the approach in realistic settings. Still, considering the novelty of the problem, the soundness of the analysis, and the fact that all reviewers leaned toward acceptance despite their reservations, I believe this work makes a valuable theoretical contribution. I recommend acceptance.